# Deep autoregressive density nets vs neural ensembles for model-based offline reinforcement learning

## Abstract

We consider the problem of offline reinforcement learning where only a set of system transitions is made available for policy optimization. Following recent advances in the field, we consider a model-based reinforcement learning algorithm that infers the system dynamics from the available data and performs policy optimization on imaginary model rollouts. This approach is vulnerable to exploiting model errors which can lead to catastrophic failures on the real system. The standard solution is to rely on ensembles for uncertainty heuristics and to avoid exploiting the model where it is too uncertain. We challenge the popular belief that we must resort to ensembles by showing that better performance can be obtained with a single well-calibrated autoregressive model on the D4RL benchmark. We also analyze static metrics of model-learning and conclude on the important model properties for the final performance of the agent.

## 1 Introduction

Reinforcement learning consists in learning a control agent (policy) by interacting with a dynamical system (environment) and collecting its feedback (rewards). This learning paradigm turned out to be able to solve some of the world's most difficult problems (Silver et al., 2017; 2018; Mnih et al., 2015; Vinyals et al., 2019). However, the scope of the systems that RL is capable of solving remains restricted to the simulated world and does not extend to real engineering systems. Two of the main reasons are i) small data due to operational constraints and ii) safety standards of such systems. In an attempt to bridge the gap between RL and engineering systems, we motivate the setting of offline reinforcement learning (Levine et al., 2020).

Offline reinforcement learning removes the need to query a dynamical system by using a previously collected dataset of controller-system interactions. In this optic, we view this setting as a supervised learning problem where one tries to approximate the underlying distribution of the data at hand, and hopefully be able to generalize to out-of-distribution samples. This turns out to be a difficult task for classical RL algorithms because of the distribution shift that occurs between the dataset and the learned policy during the learning process (Fujimoto et al., 2019; Levine et al., 2020). Thus we need to design algorithms that are well-suited for offline reinforcement learning. A common idea in this field is *conservatism* where one would only consider the learned agent when the input states are close to the support of the offline dataset. Depending on the algorithm, conservatism can take multiple forms, ranging from penalized Q-targets (Kumar et al., 2020) to uncertainty-penalized Markov decision processes (Kidambi et al., 2020; Yu et al., 2020). To develop further into this direction, we make the distinction between model-free and model-based RL (MBRL) algorithms.

Model-free algorithms learn a policy and/or a value function by observing the reward signal realizations and the underlying dynamics of the system, which in most environments requires a significant number of interactions for achieving good performance (Haarnoja et al., 2018). In this category, a way to incorporate conservatism is to penalize the value targets of data points that are distant from the offline dataset (Kumar et al., 2020). Other methods include behavior regularized policy optimization (Wu et al., 2020).

Model-based algorithms are composed of two independent (and often alternating) steps: i) model learning: a supervised learning problem of learning the dynamics (and sometimes also the reward

function) of the system of interest; and ii) policy optimization, where we sample from the learned dynamics to learn a policy and/or a value function. MBRL is known to be sample-efficient, since policy/value learning is done (completely or partially) from imaginary model rollouts (also called background planning) that are cheaper and more accessible than rollouts in the true dynamics (Janner et al., 2019). Furthermore, a predictive model with good out-of-distribution performance affords easy transfer of the true model to new tasks or areas not covered in the offline dataset (Yu et al., 2020). Conservatism in MBRL is frequently achieved by uncertainty-based penalization of the model predictions. This relies on well-calibrated estimation of the epistemic uncertainty of the learned dynamics, which is a limitation of this approach.

It is of great interest to build models that know when (and how much) they do not know, thus uncertainty estimation remains a central problem in MBRL. Many recent works have made progress in this direction (Osband et al., 2021). The most common approach to date is *bootstrap ensembles*: we construct a population of predictive models (most often probabilistic neural networks) and consider disagreement metrics as our uncertainty measurement. The source of randomness in this case is the random initialization of the parameters of neural networks and the subset of the training data that each model sees. When the environment is stochastic, ensembles help to separate the aleatory uncertainty (intrinsic randomness of the environment) and the epistemic uncertainty (Chua et al., 2018). When the environment is deterministic (which is the case of the D4RL Mujoco benchmark environments considered in most of the offline RL literature (Fu et al., 2021a)), the error is fully epistemic: it consists of the estimation error (due to lack of training data) and the approximation error (mismatch between the model class and the true distribution) (Hüllermeier & Waegeman, 2021). This highlights the need of well-calibrated probabilistic models whose posterior variance can be used as an uncertainty measurement in conservative MBRL.

In this work, we propose to compare autoregressive dynamics models (Uria et al., 2016) to ensembles of probabilistic feedforward models, both in terms of static evaluation (supervised learning metrics on the task of learning the system dynamics) and dynamic evaluation (final performance of the MBRL agent that uses the model). Autoregressive models learn a conditional distribution of each dimension of the next state conditioned on the input of the model (current state and action) and the previously generated dimensions of the next state. Meanwhile, probabilistic feedforward models learn a multivariate distribution of the next state conditioned on the current state and action. We argue that autoregressive models can learn the implicit functional dependence between state dimensions, which makes them well-calibrated, leading to good uncertainty estimates suitable for conservatism in MBRL.

Our key contributions are the following.

- We apply autoregressive dynamics models in the context of offline model-based reinforcement learning and show that they improve over neural ensembles in terms of static evaluation metrics and the final performance of the agent.

- We introduce an experimental setup that decouples model selection from agent selection to reduce the burden of hyperparameter optimization in offline RL.

- We study the impact of static metrics on the dynamic performance of the agents, and conclude on the importance of single-step calibratedness in model-based offline RL.

## 2 RELATED WORK

Offline RL has been an active area of research following its numerous applications in domains such as robotics (Chebotar et al., 2021), healthcare (Gottesman et al., 2018), recommendation systems (Strehl et al., 2010), and autonomous driving (Kiran et al., 2022). Despite outstanding advances in online RL (Haarnoja et al., 2018; Silver et al., 2017; Mnih et al., 2015) and iterated offline RL (Wang et al., 2019; Wang & Ba, 2020; Matsushima et al., 2021; Kégl et al., 2021), offline RL remained a challenging problem due to the dependency on the data collection procedure and its potential lack of exploration (Levine et al., 2020).

Although any off-policy model-free RL agent can theoretically be applied to offline RL (Haarnoja et al., 2018; Degris et al., 2012; Lillicrap et al., 2016; Munos et al., 2016), it has been shown that these algorithms suffer from distribution shift and yield poor performance (Fujimoto et al., 2019;

Levine et al., 2020). To alleviate the problem of distribution shift, *conservatism* was introduced successfully by several techniques, such as BEAR (Kumar et al., 2019), AlgaeDICE (Nachum et al., 2019), AWR (Peng et al., 2020), BRAC (Wu et al., 2020), and CQL Kumar et al. (2020). The general objective of these methods is to keep the model-free policy close to the behavioral policy, in other words, to avoid wandering into regions of the state/action space where no data was collected.

Model-based RL has been successfully applied to the online RL setting by alternating model learning and planning (Deisenroth & Rasmussen, 2011; Hafner et al., 2021; Gal et al., 2016; Levine & Koltun, 2013; Chua et al., 2018; Janner et al., 2019; Kégl et al., 2021). Planning is done either decision-time via model-predictive control (Draeger et al., 1995; Chua et al., 2018; Hafner et al., 2019; Pinneri et al., 2020; Kégl et al., 2021)), or Dyna style by learning a model-free RL agent on imagined model rollouts (Janner et al., 2019; Sutton, 1991; Sutton et al., 1992; Ha & Schmidhuber, 2018). For instance, MBPO (Janner et al., 2019) trains an ensemble of feed-forward models and generates imaginary rollouts to train a soft actor-critic, which policy is then used to generate new data for model learning. MBPO has been showed to achieve state of the art in continuous control task with the smallest sample efficiency. An adaptation of MBPO to the offline setting is MOPO (Yu et al., 2020). MOPO incorporates conservatism via a surrogate MDP where the rewards are penalized with the uncertainty of the model. While MOPO relies on disagreement metrics between the members of the learned ensemble, we suggest the use of well-calibrated autoregressive models whose learned variance is a good proxy to the model estimation error. Similar uncertainty penalized policy search is used in a number of other works (Kidambi et al., 2020; Lee et al., 2021; Shen et al., 2021; Swazinna et al., 2021; Depeweg et al., 2018), while others explore pessimism-based decision time planning (Argenson & Dulac-Arnold, 2021; Zhan et al., 2021), conservative value learning (Yu et al., 2021; Liu et al., 2021).

Autoregressive models have been studied in a number of previous works for generative modeling in general (Uria et al., 2016; 2013; Papamakarios et al., 2017; Van Den Oord et al., 2016). However, only a handful of papers use them in the context of MBRL (Kégl et al., 2021; Zhang et al., 2021b; Zhan et al., 2021). Zhang et al. (2021b) used autoregressive models for model-based off-policy evaluation, while we focus our study on the important model properties for offline policy optimization. We also adapt metrics from Kégl et al. (2021) to provide a complete guide on model selection for offline MBRL.

Previous works have tackled hyperparameter selection in online RL (Andrychowicz et al., 2021; Engstrom et al., 2020), MBRL (Zhang et al., 2021a), and offline RL (Paine et al., 2020), showing the sensibility of existing algorithms to hyperparameter choices. Lu et al. (2022) perform a similar analysis to this work. Similarly to us, they base their analysis on MOPO, but they focus on the uncertainty-related hyperparameters while we revisit the model design and architecture.

## 3 PRELIMINARIES

The standard framework of RL is the finite-horizon **Markov decision process (MDP)** $\mathcal{M} = \langle \mathcal{S}, \mathcal{A}, p, r, \mu_0, \gamma \rangle$ where $\mathcal{S}$ represents the state space, $\mathcal{A}$ the action space, $p : \mathcal{S} \times \mathcal{A} \rightsquigarrow \mathcal{S}$ the (possibly stochastic) transition dynamics, $r : \mathcal{S} \times \mathcal{A} \rightarrow \mathbb{R}$ the reward function, $\mu_0$ the initial state distribution, and $\gamma \in [0, 1]$ the discount factor. The goal of RL is to find, for each state $s \in \mathcal{S}$, a distribution $\pi(s)$ over the action space $\mathcal{A}$, called the *policy*, that maximizes the expected sum of discounted rewards $J(\pi, \mathcal{M}) := \mathbb{E}_{s_0 \sim \mu_0, a_t \sim \pi, s_{t>0} \sim p}[\sum_{t=0}^{H} \gamma^t r(s_t, a_t)]$, where $H$ is the MDP horizon. Under a policy $\pi$, we define the state-action value function (*Q-function*) at an $(s, a) \in \mathcal{S} \times \mathcal{A}$ pair as the expected sum of discounted rewards, starting from the state $s$, taking the action $a$, and following the policy $\pi$ afterwards until termination: $Q^\pi(s, a) = \mathbb{E}_{a_{t>0} \sim \pi, s_{t>0} \sim p}\left[\sum_{t=0}^{H} \gamma^t r(s_t, a_t) \mid s_0 = s, a_0 = a\right]$. We can similarly define the state value function by taking the expectation with respect to the initial action $a_0$: $V^\pi(s) = \mathbb{E}_{a_t \sim \pi, s_{t>0} \sim p}\left[\sum_{t=0}^{H} \gamma^t r(s_t, a_t) \mid s_0 = s\right]$.

In **offline RL**, we are given a set of transitions $\mathcal{D} = \{(s_t^i, a_t^i, r_t^i, s_{t+1}^i)\}_{i=1}^{N}$, where $N$ is the size of the set, generated by an unknown behavioral policy $\pi^\beta$. The difficulty of offline RL comes from the fact that we are not allowed to interact further with the environment $\mathcal{M}$ even though we aim to optimize the objective $J(\pi, \mathcal{M})$ with $\pi \neq \pi^\beta$. In practice, the current offline RL algorithms are still provided with an online evaluation budget, a setting we will follow in the rest of the paper.

The question of offline policy evaluation (or budget-limited policy evaluation) is an active research direction (see, e.g., Fu et al. (2021b)) and is beyond the scope of this paper.

**Model-based RL** algorithms use an offline dataset $\mathcal{D}$ to solve the supervised learning problem of estimating the dynamics of the environment $\hat{p}$ and/or the reward function $\hat{r}$. For various reasons (stochastic environment, ability to represent the uncertainty of the predictions), the loss function is usually the log-likelihood $\mathcal{L}(\mathcal{D}; \hat{p}) = \frac{1}{N} \sum_{i=1}^{N} \log \hat{p}(s_{t+1}^i | s_t^i, a_t^i)$. The learned model can then be used for policy search under the MDP $\hat{\mathcal{M}} = \langle \mathcal{S}, \mathcal{A}, \hat{p}, \hat{r}, \mu_0, \gamma \rangle$, which has the same state and action spaces $\mathcal{S}, \mathcal{A}$ as the true environment $\mathcal{M}$, but which has the transition probability $\hat{p}$ and the reward function $\hat{r}$ that are learned from the offline data $\mathcal{D}$. The obtained optimal policy $\hat{\pi} = \text{argmax}_{\pi} J(\pi, \hat{\mathcal{M}})$ is not guaranteed to be optimal under the true MDP $\mathcal{M}$ due to distribution shift and model bias. $J(\pi, \hat{\mathcal{M}})$ and $J(\pi, \mathcal{M})$ are somewhat analogous to training and test scores in supervised learning, with two fundamental differences: i) they are only loosely connected to the actual supervised loss $\mathcal{L}(\mathcal{D}; \hat{p})$ that we can optimize and measure on a data set, and ii) because we are not allowed to collect data using $\pi$, there is a distribution shift between training and test.

Regarding the type of model, the usual choice is a probabilistic model that learns the parameters of a multivariate Gaussian over the next state and reward, conditioned on the current state and action: $s_{t+1}, r_t \sim \hat{p}_{\theta}(.|s_t, a_t) = \mathcal{N}\big(\mu_{\theta}(s_t, a_t), \sigma_{\theta}(s_t, a_t)\big)$, where $\theta$ represents the parameters of the predictive model. In practice, we use fully connected neural networks as they are proved to be powerful function approximators (Nagabandi et al., 2018; Chua et al., 2018), and for their suitability to high-dimensional environments over simpler non-parametric models such as Gaussian processes. Following previous work (Chua et al., 2018), we assume a diagonal covariance matrix for which we learn the logarithm of the diagonal entries: $\sigma_{\theta} = \text{Diag}(\exp(l_{\theta}))$ with $l_{\theta}$ output by the neural network.

One of the conditions of such a joint model is the conditional independence of the dimensions of the predicted state, which is a strong assumption, especially in the case of functional (or physical) dependency. $y$-interdependence (Kégl et al., 2021) happens, for example, when angles are represented by sine and cosine. For this purpose, we study **autoregressive** models that learn a single model per dimension, conditioned on the input of the model $(s_t, a_t)$ *and* the previously generated dimensions. Formally, $\hat{p}_{\theta}(s_{t+1}|s_t, a_t) = \hat{p}_{\theta_1}(s_{t+1}^1|s_t, a_t) \prod_{j=2}^{d_s} \hat{p}_{\theta_j}(s_{t+1}^j|s_{t+1}^1, \ldots, s_{t+1}^{j-1}, s_t, a_t)$, where $d_s$ is the dimension of the state space $\mathcal{S}$.

Conservatism in MBRL requires an uncertainty estimate $\hat{u}(s, a)$ reflecting the quality of the model in different regions of the state/action space. For this purpose, probabilistic models provide an uncertainty estimate by learning the variance of the predictions (in this case under a Gaussian distribution). In noisy environments, this uncertainty estimate represents both the *aleatory* uncertainty (intrinsic randomness of the environment) and the *epistemic* uncertainty (model estimation and approximation errors). Conservative MBRL uses the epistemic uncertainty only, so, in practice, the problem of separating the aleatory uncertainty and the epistemic uncertainty is addressed through the use of bootstrap ensembles (Chua et al., 2018). Ensembling consists in having $D \in \mathbb{N}^* - \{1\}$ models, each initialized randomly and trained on a set $\mathcal{D}_\ell$ for $\ell \in \{1, \ldots, D\}$ generated by sampling with replacement from a common dataset $\mathcal{D}$. Using ensembles, we can compute a disagreement metric to capture the epistemic uncertainty, as opposed to the aleatory uncertainty learned by each member of the ensemble. A detailed discussion about these uncertainty heuristics is provided in Section 4.

## 4    A BASELINE: MODEL-BASED OFFLINE POLICY OPTIMIZATION (MOPO)

Models $\hat{p}$ in MBRL are not used in isolation. Their likelihood ratio, precision, and calibratedness (LR, R2, and KS in Section 5.1 and Appendix C) are good proxies, but ultimately their quality is judged when they are used in a policy. To compare the dynamic performance of the models $\hat{p}$, we fix the policy to MOPO (Yu et al., 2020), a conservative agent-learning algorithm. MOPO uses a pessimistic MDP (P-MDP) to ensure that the performance of the policy with the model will be a lower bound of the performance of the policy on the real system. Yu et al. (2020) show a theoretical lower bound on the true return based on the estimation error of the learned dynamics $J(\pi, \mathcal{M}) \geq \mathbb{E}_{a \sim \pi, s \sim \hat{p}} \big[ r(s, a) - \gamma |G_{\hat{\mathcal{M}}}^{\pi}(s, a)| \big]$. In this formula, $G_{\hat{\mathcal{M}}}^{\pi}(s, a)$ is defined by $\mathbb{E}_{s' \sim \hat{p}(s,a)}[V_{\mathcal{M}}^{\pi}(s')] - \mathbb{E}_{s' \sim p(s,a)}[V_{\mathcal{M}}^{\pi}(s')]$ which quantifies the effect of the model error on the return. However, this

requires access to the value function of the policy $\pi$ under the true MDP $\mathcal{M}$, which is not given in practice.

To derive an algorithm based on this theoretical bound, MOPO relies on an upper bound of $G_{\tilde{\mathcal{M}}}^\pi(s, a)$ based on the integral probability metric: $G_{\tilde{\mathcal{M}}}^\pi(s, a) \leq \sup_{f \in \mathcal{F}} \mid \mathbb{E}_{s' \sim \hat{p}}[f(s')] - \mathbb{E}_{s' \sim p}[f(s')] \mid$, where $\mathcal{F}$ is an arbitrary set of functions. In practice, the authors use ensemble-based uncertainty heuristics to set an upper bound on the true error of the model. The maximum standard deviation among the ensemble members (labeled **max aleatory or MA**) is considered to define a penalized reward $\tilde{r}(s, a) = \hat{r}(s, a) - \lambda \hat{u}(s, a)$, where $\hat{u}(s, a) = \max_{\ell=1,\ldots,N} \|\sigma_\theta^\ell(s, a)\|_F$ and $\lambda$ is a penalty hyperparameter. Yu et al. (2020) then define the associated P-MDP $\tilde{\mathcal{M}} = \langle \mathcal{S}, \mathcal{A}, \hat{p}, \tilde{r}, \mu_0, \gamma \rangle$ on which a soft actor-critic (SAC) (Haarnoja et al., 2018) agent is trained until convergence (Algorithm 1). This algorithm is based on Model-based policy optimization (MBPO) (Janner et al., 2019) which alternates between model learning and agent learning. MOPO can be described as one iteration of MBPO, which learns the dynamics model (a bootstrap ensemble of probabilistic neural networks) from the offline dataset and then learns the off-policy agent on a buffer[1] of rollouts in the P-MDP $\tilde{\mathcal{M}}$. Using this P-MDP prevents the agent from exploiting rewards of highly uncertain regions.

---

**Data:** Dataset $\mathcal{D}$, penalty coefficient $\lambda$, rollout horizon $h$, Number of SAC training batches $B$, conservatism penalty $\hat{u}(s, a)$.

Train dynamics model $\hat{p}$ on offline dataset $\mathcal{D}$;
Initialize SAC policy $\pi$ and empty replay buffer $\mathcal{D}_{model}$;
**for** $1, 2, \ldots, B$ **do**
    Sample initial state $s_0$ from $\mathcal{D}$;
    **for** $i = 1, 2, \ldots, h$ **do**
        Sample an action $a_i \sim \pi(s_i)$;
        Sample the next state from the dynamics model $s_{i+1}, r_i \sim \hat{p}(s_i, a_i)$;
        Compute the penalized reward $\tilde{r}_i = r_i - \lambda \hat{u}(s_i, a_i)$;
        Add sample $(s_i, a_i, \tilde{r}_i, s_{i+1})$ to $\mathcal{D}_{model}$;
    **end**
    Draw a batch from $\mathcal{D}_{model}$, update $\pi$ following SAC schema;
**end**

**Algorithm 1:** MOPO pseudocode. Yu et al. (2020) uses $\hat{u}(s, a) = \max_{\ell=1,\ldots,N} \|\sigma_\theta^\ell(s, a)\|_F$; we also experimented with two other penalty heuristics by Lu et al. (2022).

---

While Yu et al. (2020) only tried the max aleatory estimator for the uncertainty heuristic, Lu et al. (2022) introduced concurrent ensemble-based uncertainty heuristics from recent works and deployed them in MOPO. Among these, we chose the following two, showing competitive performance in benchmarks.

- **Max pairwise difference (MPD)** (Kidambi et al., 2020): $\hat{u}(s, a) = \max_{l,l'} \|\mu_{\theta_l}^l(s, a) - \mu_{\theta_{l'}}^{l'}(s, a)\|_2$ for $l \neq l' \in 1, \ldots, D$. This metric captures the largest disagreement among ensemble members as an indicator of model error.

- **Ensemble standard deviation (ESD)** (Lakshminarayanan et al., 2017):
  $\hat{u}(s, a) = \sqrt{\frac{1}{D} \sum_l^D \left( (\sigma_{\theta_l}^l(s, a))^2 + (\mu_{\theta_l}^l(s, a))^2 \right) - (\bar{\mu}(s, a))^2}$ with $\bar{\mu}(s, a) = \frac{1}{D} \sum_l^D \mu_{\theta_l}^l(s, a)$, is the standard deviation of the ensemble, i.e., the standard deviation of the equally-weighted mixture of the Gaussian densities.

## 5 EXPERIMENTAL SETUP

We implement our MOPO baseline based on the MBRL library released by Kégl et al. (2021) which is built on top of the RAMP framework (Kégl et al., 2018). We run our experiments with the following models:

---

[1] Initially, MOPO selected 5% of the batch from the real system dataset $\mathcal{D}$, and 95% of model rollouts. However, Yu et al. (2020) show that this does not influence the performance of the algorithm.

- DARMDN($D$): Deep autoregressive mixture density net. $d_s \in \mathbb{N}^*$ feed-forward neural network that learn the parameters (mean and log-standard deviation), and the weights of $D \in \mathbb{N}^*$ univariate Gaussian distributions ($d_s$ being the dimension of the state space $\mathcal{S}$). Although our implementation is general, for the rest of the paper we only consider DARMDN(1) due to runtime bottleneck, we refer to it as simply DARMDN.

- DMDN($D$): Deep mixture density net. A feed-forward neural network that learns the parameters (mean and log-standard deviation) and the weights of $D \in \mathbb{N}^*$ multivariate Gaussian distributions. For similar reasons as DARMDN, we only consider DMDN(1) and refer to it as DMDN.

- ENS: Ensemble of $D \in \mathbb{N}^*$ DMDN models. We implement a vectorized version that is optimized to run on a Graphical Processing Units (GPUs). Notice that ENS is equivalent to the original model MOPO used, modulo architectural choices.

For the single models (DARMDN and DMDN), we consider their learned standard deviation ($\sigma_\theta$) as the uncertainty heuristic to use for reward penalization, which is equivalent to the max aleatory heuristic for an ensemble of a single member. For ENS, we follow the schema by Lu et al. (2022) and tune the uncertainty heuristic as an additional hyperparameter among MA, MPD, and ESD, defined in Section 4.

In a typical MBRL loop, the experimental setup consists of alternating model learning and agent learning until the potential convergence of the dynamic performance (episodic return) of the agent on the real environment. For computationally limited hyperparameter optimization, this setup provides continuous feedback on the return of a given model, which helps to early-stop unpromising experiments. This is not possible in single-iteration offline RL as we only have access to a static dataset for model learning, and we have to run all the pipeline to compute the evaluation score of a given model. For this purpose, we suggest to decouple model selection and agent selection in an attempt to reduce the overall computational budget of the approach. The experimental setup will then be separated to two independent parts:

- **Static evaluation of the models:** Starting from a dataset $\mathcal{D}$, we evaluate the different models by computing supervised-learning evaluation metrics (Sections 5.1 and C) on a held-out validation set. We then select the best model hyperparameters based on these metrics.

- **Dynamic evaluation of the agents:** After selecting the best model $\hat{p}$, we train agents by interacting with the P-MDP defined on the learned dynamics of the model. During training, we evaluate the agents by repeatedly rolling-out trajectories in the real environment and computing their average episodic return. For this purpose, we assume access to the true simulator at evaluation time, although the recorded episodes are not made available to training.

A limitation of this approach comes from the fact that static supervised learning metrics do not necessarily reflect the quality of the model for agent learning. We thus investigate how these static metrics predict the overall dynamic performance in Section 6.

## 5.1 STATIC METRICS

We use metrics introduced by Kégl et al. (2021) in the context of iterated offline reinforcement learning. These metrics are designed to assess different aspects of model quality: precision, calibratedness, and sensitivity to compounding errors via long-horizon metrics.

Precision is evaluated using the explained variance (**R2**) which we prefer over the standard Mean-squared error (MSE) because it is normalized and can be aggregated over multiple dimensions. Calibratedness is measured using the Kolmogorov-Smirnov statistics (**KS**) between the ground truth validation quantiles and a uniform distribution. This metrics indicates if the ground truth values are distributed following the predicted distributions. In the Gaussian case, it is equivalent to the predicted standard deviation being in the order of magnitude of the true model error (although a bad KS may also indicate that the model errors are not Gaussian). We also use the likelihood ratio with respect to a baseline score (**LR**), and the outlier ratio (**OR**), the rate of data points on which the likelihood is close to zero. For the impact of compounding errors, we sample a population of

trajectories (following ground truth actions) and compute Monte-Carlo estimates of the long-horizon metrics (**R2(L)** and **KS(L)** for $L \in \{1, \ldots, 20\}$). The formal definition of these metrics can be found in Appendix C.

## 5.2 DYNAMIC METRICS

Similarly to Kidambi et al. (2020); Wu et al. (2020), we compute the average episodic return (undiscounted sum of rewards) of the agent on the real system during training, formally $R(\{(s_t, a_t, r_t, s_{t+1})\}_{t=1}^H) = \sum_{t=1}^H r_t$ of the agent, where $H$ is the episode size. We then keep track of the agent with the highest return for the final evaluation. This is not what we should do if the goal was to develop a standalone offline RL algorithm (we could not use the real return to select the agent), but our goal in this paper is to compare models $\hat{p}$ of the system dynamics, so as long as the agent is selected in the same way for all the models, the comparison is fair.[1]

We use the normalized scores introduced in the D4RL benchmark. This metric is a linear transformation of the episodic return and takes values between 0 and 100 with 0 corresponding to the score of a randomly initialized SAC agent, and 100 to a SAC agent that is trained until convergence on the real system.

## 6 EXPERIMENTS & RESULTS

All our experiments are conducted in the continuous control environment *Hopper*. We use the implementation of OpenAI Gym (Brockman et al., 2016) that is based on the Mujoco physics simulator (Todorov et al., 2012). A description of this environment can be found in Appendix B.

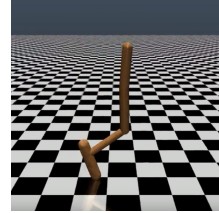

Figure 1: Hopper

For static datasets, we use the D4RL *Hopper* benchmark that provides four static sets generated by different behavior policies (*random*: 1M steps generated by a randomly initialized SAC agent, *medium*: 1M steps generated by a SAC agent trained until half the score at convergence, *medium-replay*: All the traces collected by a SAC agent trained until half the score at convergence, *medium-expert*: 2M steps consisting of the *medium* dataset and 1M steps generated by an expert SAC agent).

The results of the static evaluation of the models are summarized in Table 1. The reported scores are validation scores on a held-out 10% validation set from the D4RL datasets.

Table 1: Model evaluation results on static datasets. $\downarrow$ and $\uparrow$ mean lower and higher the better, respectively. Unit is given after the / sign.

| Model | LR$\uparrow$ | OR/$10^{-4}\downarrow$ | R2/$10^{-4}\uparrow$ | KS/$10^{-3}\downarrow$ | R2(10)/$10^{-4}\uparrow$ | KS(10)/$10^{-3}\downarrow$ |
|---|---|---|---|---|---|---|
| | | | Hopper-v2, D4RL random dataset | | | |
| DMDN | 976 | 0 | 9986 | 146 | 8017 | 190 |
| DARMDN | 1141 | 0 | 9987 | 134 | 5011 | 377 |
| ENS | 304 | 1 | 9984 | 217 | 9442 | 190 |
| | | | Hopper-v2, D4RL medium dataset | | | |
| DMDN | 1322 | 1 | 9998 | 117 | 9938 | 84 |
| DARMDN | 1473 | 2 | 9996 | 56 | 9586 | 112 |
| ENS | 341 | 1 | 9953 | 233 | 9296 | 143 |
| | | | Hopper-v2, D4RL medium-replay dataset | | | |
| DMDN | 361 | 3 | 9990 | 120 | 9817 | 141 |
| DARMDN | 575 | 4 | 9986 | 65 | 9580 | 141 |
| ENS | 219 | 1 | 9928 | 190 | 6982 | 115 |
| | | | Hopper-v2, D4RL medium-expert dataset | | | |
| DMDN[2] | - | - | - | - | - | - |
| DARMDN | 1675 | 1 | 9996 | 59 | 8814 | 160 |
| ENS | 452 | 1 | 9976 | 227 | 9727 | 108 |

---

[1]As a related remark, we consider the giant variance of the return both across seeds and across training iterations of the agent crucial, arguably the most important problem of offline RL, but outside the scope of this paper.

**One-step metrics (LR, OR, R2, and KS).** We first observed that single models are consistently better than the ensemble in terms of one-step metrics. To better understand this result, we propose to use the ground truth test quantiles as a debugging tool on the calibratedness of the models. Figure 2 and Appendix E show that the ensemble model overestimates its error because the ground truth values are concentrated around 0.5. We suggest that this is because each DMDN ensemble member has a well-calibrated variance, but when we treat the ensemble as a mixture model, the variance of the mean adds to the individual variances, "diluting" the uncertainty. Regarding the comparison between DMDN and the autoregressive DARMDN, we observe that although they have similar R2 scores, DARMDN is consistently beating DMDN in terms of KS and LR which depend also on accurate and well-calibrated sigma estimates, an important property for conservative MBRL.

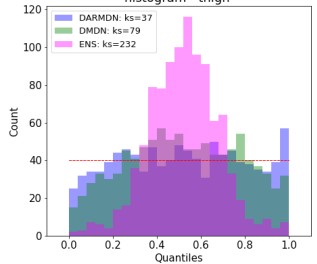

Figure 2: Histogram of Hopper's *thigh* ground truth quantiles, under the model distribution (D4RL medium dataset). The legend also includes the value of the KS calibratedness metric. The dotted red line indicates the ideal case when the quantiles follow a uniform distribution.

To push the analysis further, we suggest to look at the dimension-wise static metrics, reported in Appendix D. The results depend on the different datasets, yet some results are consistent and help explain the improvement that autoregressive models bring over their counterparts. For instance in three out of four datasets, the LR score of the *thigh* and *thigh_dot* dimensions is an order of magnitude higher for the autoregressive model. We suggest that this is due to the functional dependence that might exist between the different observables, which is easily captured by the autoregressive model as it uses the previously predicted dimensions as input to the next ones.

**Long-horizon metrics.** Unlike in single-step metrics, here we observe a significant degradation in the performance of DARMDN, both in terms of $R2(L)$ and $KS(L)$ for $L \in \{1, \ldots, 20\}$ (Figure 3 and Appendix F). We suggest that this is the due to optimizing the models for single-step likelihood. Outliers (last bin of Figure 2) count little in the single-step likelihood, but may compound when recursing the model through $L$ steps.

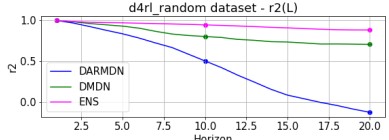

Figure 3: Long horizon explained variance $\mathbf{R2}(L)$ in the D4RL random dataset.

**Dynamic evaluation.** Table 2 shows the episodic return achieved by the best agent throughout one million steps of SAC training. SAC agents that were trained using DARMDN models performed better on the real system despite their suboptimal long-horizon performance. We suggest that for an agent that trained by one-step Q-learning, such as SAC, only one-step errors matter. Ensemble models improve over DMDN in the random dataset, but scores are comparable or worse in the remaining tasks, although none of the differences are highly significant (they depend on a couple of lucky seeds; a phenomenon that muddies the offline RL field). One result seems remarkable: DARMDN models seem to be able to consistently generate agents that go beyond Hopper simply standing up (score of about 30).

Table 2: Model dynamic evaluation: mean $\pm$ std over 3 seeds of the hyperoptimal SAC agents. The reported score is the D4RL normalized score explained in Section 5.

| D4RL dataset | DARMDN | DMDN | ENS |
|---|---|---|---|
| random | 31.34 $\pm$ 0.50 | 17.54 $\pm$ 9.80 | 31.97 $\pm$ 0.26 |
| medium | 66.96 $\pm$ 6.33 | 33.12 $\pm$ 5.96 | 14.12 $\pm$ 13.25 |
| medium_replay | 70.57 $\pm$ 24.45 | 33.16 $\pm$ 1.90 | 28.96 $\pm$ 15.87 |
| medium_expert | 64.18 $\pm$ 25.10 | - $\pm$ - | 32.10 $\pm$ 0.67 |

**Correlating static metrics and dynamic scores.** The experimental setup we introduce has the advantage of reducing the combinatorics of the hyperparameter optimization process. However, the best agents do not necessarily come from the models with the best static metrics, since these are

---

[2]The medium-expert dataset contains 2M timesteps which is costly in compute and memory. We therefore, omit this experiment.

measured on static data not representative of the distribution on which they are applied in the dynamic run. In an attempt to optimize model selection, we investigate the model properties (static metrics) that are most important for dynamic scores. For this, we compute Spearman rank correlation ($\rho$) and Pearson bivariate correlation ($r$) between the static score obtained for all models and their respective dynamic scores.

A value of $\rho = 1$ indicates that the static metric conserves the same ranking observed in the dynamic evaluation (sufficient for model selection) while $r = 1$ tells that the gap observed in the static metric is in the same scale of the one observed in the dynamic performance (linear correlation). The results in Figure 4 and Appendix G show that in most datasets, the two most correlating metrics are LR ($\rho = 1.0$ and $r = 0.93$) and KS(1) ($\rho = 1.0$ and $r = 0.83$), metrics that evaluate the calibratedness of the mod-

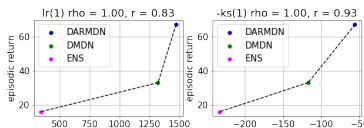

Figure 4: The Spearman and Pearson correlations between the episodic return and **LR/-KS** metrics on the D4RL medium dataset.

els. This underlines the fact that autoregressive models yield the best agent because of their ability to learn one-step uncertainty estimates that represent well their true errors.

**D4RL benchmark.** We compare the scores obtained with our best agent (based on an autoregressive model) with existing literature in the D4RL benchmark and include the results in Table 3.

Table 3: Results on the D4RL benchmark. The scores indicate the mean $\pm$ standard deviation across 3 seeds (6 seeds for MOPO) of the normalized episodic return. We take the scores of MBRL algorithms from their respective papers, and the scores of the model free algorithms and Behavior cloning (BC) from the D4RL paper (Fu et al., 2021a).

| D4RL dataset | BC | Ours | MOPO | COMBO | MOREL | SAC | BEAR | BRAC-v | CQL |
|---|---|---|---|---|---|---|---|---|---|
| random | 1.6 | $31.3 \pm 0.5$ | $11.7 \pm 0.4$ | 17.9 | 53.6 | 11.3 | 9.5 | 12.0 | 10.8 |
| medium | 29.0 | $66.9 \pm 6.3$ | $28.0 \pm 12.4$ | 94.9 | 95.4 | 0.8 | 47.6 | 32.3 | 86.6 |
| medium_replay | 11.8 | $70.5 \pm 24.4$ | $67.5 \pm 24.7$ | 73.1 | 93.6 | 1.9 | 10.8 | 0.9 | 48.6 |
| medium_expert | 111.9 | $64.1 \pm 25.1$ | $23.7 \pm 6.0$ | 111.1 | 108.7 | 1.6 | 4.0 | 0.8 | 111.0 |

Our algorithm achieves better or similar (*medium_replay*) performance than MOPO, suggesting that potentially the improvement is brought by autoregressive models over neural ensembles, which supports the case of single well-calibrated models. However, we would like to emphasize that there may be other potential reasons behind such differences of performance. For instance Kidambi et al. (2020) append the observations with the unobserved *x_velocity* to get access to the full state of the true MDP. The D4RL dataset version (*v0* or *v2*) has also been criticized as providing different qualities for the same dataset (we use *v2* similar to Kidambi et al. (2020) and Yu et al. (2021) while Yu et al. (2020) uses *v0*)[3]. Another important point is the evaluation protocol that sometimes assumes access to the real system for policy evaluation (Kidambi et al., 2020; Wu et al., 2020; Fujimoto et al., 2019; Kumar et al., 2019), and sometimes only reports the online evaluation score of the policy at the last agent-training iteration (Yu et al., 2020; 2021). Finally the architectural choices of the model design and the chosen policy optimization algorithm can also impact the performance. Consequently, we believe that, beyond designing benchmark data set, providing a unified *evaluation framework* for offline RL is highly necessary. We plan to explore this direction in future work.

## 7 CONCLUSION

In this paper, we ask what are the best neural networks based dynamic system models, estimating their own uncertainty, for conservativism-based MBRL algorithms. We build on a previous work by Yu et al. (2020) (MOPO: model-based offline policy optimization) who use bootstrap ensembles. Throughout a rigorous empirical study incorporating metrics that assess different aspects of the model (precision, calibratedness, long-horizon performance), we show that deep autoregressive models can improve upon the baseline in Hopper, one of the D4RL benchmark environments. Our results exhibit the importance of calibratedness when the learned variance is used as an uncertainty heuristic for reward penalization. Future work includes confirming our results on other benchmarks and designing a unified offline RL evaluation protocol.

---

[3]Some issues have been raised about this in prior work: Lu et al. (2022), Issue_1, Issue_2.

## REPRODUCIBILITY STATEMENT

In order to ensure reproducibility we will release the code at `<URL hidden for review>`, once the paper has been accepted.

Finally, the hyperparameters of the algorithms are listed in Appendix A and the pseudocode is shown in Section 4.

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

## A    IMPLEMENTATION DETAILS

**MOPO Implementation details.** Following MBPO, MOPO uses a bootstrap ensemble of proba-bilistic neural networks $\hat{p}_\theta^\ell = \mathcal{N}(\mu_\theta^\ell, \sigma_\theta^\ell)\}_{\ell=1}^D$ trained independently by log-likelihood maximization. The dynamics model is a four-layer neural network with 200 units each, swish activation functions and ridge regularization with different weight decays on each hidden layer. During the model roll-out generation phase, MOPO first samples initial states from the offline dataset, then performs short rollouts on the learned dynamics (with the horizon $h \in \{1, 5\}$).

**Our Implementation details.** For all the models, we use a neural network composed of a com-mon number of hidden layers and two output heads (with *Tanh* activation functions) for the mean and standard deviation of the learned probabilistic dynamics. We use batch normalization (Ioffe & Szegedy, 2015), Dropout layers (Srivastava et al., 2014), and set the learning rate of the Adam optimizer (Kingma & Ba, 2015), the number of common layers, and the number of hidden units as hyperparameters that we tune using the built-in hyperoptimization engine in the RAMP frame-work (Kégl et al., 2018). For the ensemble implementation, we replicate the DMDN model with the optimal hyperparameters and train them by shuffling the training set (a practical variation to boot-strapping (Chua et al., 2018; Pineda et al., 2021)). In all experiments, we use an ensemble of three models. Table 4 shows the grid search ranges for the hyperparameters of our models.

Table 4: Model hyperparameters Grid search range.

| Model | DARMDN | DMDN | |
|---|---|---|---|
| Learning rate (Lr) | $10^{-3}, 3 \times 10^{-4}$ | $10^{-3}, 3 \times 10^{-4}$ | |
| Number of hidden units (Nhu) | 50, 100, 200 | 100, 200, 500 | |
| Number of common layers (Ncl) | 1, 2 | 2, 3, 4 | |

Using the one-million-timestep D4RL data sets, we first determine the best model hyperparameters (in terms of the aggregate validation static metrics) on a subset of 50K training points (and 500K validation points), then we train the best models on 90% of the whole data sets.

For the dynamic scores, we use Ray-tune (Moritz et al., 2018) to find the optimal hyperparameters (short rollouts horizon $h \in \{1, 5, 50, 100\}$, uncertainty penalty $\lambda \in \{0.1, 1, 5, 25\}$, and uncertainty heuristic for ensembles $u \in \{$Max aleatory (MA), Max pairwise difference (MPD), Ensemble stan-dard deviation (ESD)$\}$ on each model/data pair. We use the implementation of the open-source library StableBaselines3 (Raffin et al., 2021) for the SAC agents.

We give the best hyperparameters for each model/data pair in Table 5.

Table 5: The optimal hyperparameters for all model/data pair.

| Model | Lr | Nhu | Ncl | $h$ | $\lambda$ | $u$ |
|---|---|---|---|---|---|---|
| | | | D4RL random dataset | | | |
| DMDN | $3 \times 10^{-4}$ | 500 | 3 | 5 | 0.1 | - |
| DARMDN | $10^{-3}$ | 200 | 2 | 100 | 1.0 | - |
| ENS | $3 \times 10^{-4}$ | 500 | 3 | 5 | 5 | ESD |
| | | | D4RL medium dataset | | | |
| DMDN | $3 \times 10^{-4}$ | 500 | 3 | 5 | 0.1 | - |
| DARMDN | $10^{-3}$ | 200 | 1 | 5 | 0.1 | - |
| ENS | $3 \times 10^{-4}$ | 500 | 3 | 50 | 25 | ESD |
| | | | D4RL medium-replay dataset | | | |
| DMDN | $3 \times 10^{-4}$ | 500 | 3 | 5 | 0.1 | - |
| DARMDN | $10^{-3}$ | 200 | 2 | 100 | 0.1 | - |
| ENS | $3 \times 10^{-4}$ | 500 | 3 | 50 | 5 | MPD |
| | | | D4RL medium-expert dataset | | | |
| DARMDN | $10^{-3}$ | 200 | 1 | 100 | 0.1 | - |
| ENS | $3 \times 10^{-4}$ | 200 | 3 | 5 | 0.1 | ESD |

## B    CHARACTERISTICS OF THE BENCHMARK ENVIRONMENT: HOPPER

The hopper environment consists of a robot leg with 11 observations (*rootz, rooty, thigh, leg, foot, rootx_dot, rootz_dot, rooty_dot, thigh_dot, leg_dot, foot_dot*) including the angular positions and ve-locities of the leg joints, except for the $x$ position of the root joint. The action is a control signal

applied by three actuators located in the three joints. The goal of the system is to hop forward as fast as possible (maximizing the velocity in the direction of $x$) while applying the smallest possible control (measured by $\|a_t\|_2^2$), and without falling into unhealthy states (terminal states where the position of the leg is physically unfeasible). We detail the characteristics of the environment in Table 6.

Table 6: Hopper characteristics.

| dimension of the observable space | dimension of the action space | task horizon | reward function |
|---|---|---|---|
| 11 | 3 | 1000 | $\dot{x}_t - 0.1 \times \|a_t\|_2^2 + \mathbf{1}\{\text{state is healthy}\}$ |

## C  STATIC METRICS

We define our static metrics based on the marginal one-dimensional densities of each predicted feature. For autoregressive models, these densities are learned separately while non-autoregressive models learn a multivariate density that is separated using the product rule:

$$p(s_{t+1}|s_t, a_t) = p_1(s_{t+1}^1|\boldsymbol{x}_t^1) \prod_{j=2}^{d_s} p_j(s_{t+1}^j|\boldsymbol{x}_t^j) \quad \text{where} \quad \boldsymbol{x}_t^j = \left(s_{t+1}^1, \ldots, s_{t+1}^{j-1}, s_t, a_t\right).$$

All metrics will be evaluated on a data set $\mathcal{D}$ of size $N$, consisting of transitions in the real system. $\mathcal{D}$ stands for a held-out validation set on the offline training datasets.

**EXPLAINED VARIANCE (R2)**: Measures the precision of the mean predictions.

$$\text{R2}(\mathcal{D}; \theta; j \in \{1, \ldots, d_s\}) = 1 - \frac{\frac{1}{N}\sum_{i=1}^{N}\left(s_{i,t+1}^j - \mu_\theta^j(s_{i,t}^j, a_{i,t})\right)^2}{\frac{1}{N}\sum_{i=1}^{N}\left(s_{i,t+1}^j - \bar{s}_{t+1}^j\right)^2} \tag{1}$$

where $\theta$ are the model parameters and $\bar{s}_{t+1}^j$ the sample mean of the $j^{\text{th}}$ dimension of $s_{t+1}$. R2 is between 0 and 1, the higher the better.

**LIKELIHOOD RATIO (LR)**: The average log-likelihood evaluated on $\mathcal{D}$ is defined as

$$\mathcal{L}(\mathcal{D}; \theta; j \in \{1, \ldots, d_s\}) = \frac{1}{N}\sum_{i=1}^{N} \log p_\theta^j\left(s_{i,t+1}|s_{i,t}, a_{i,t}\right) \tag{2}$$

where $p_\theta$ is the PDF of the Gaussian distribution induced by the learned parameters: $\mathcal{N}\left(\mu_\theta(s_t, a_t), \sigma_\theta(s_t, a_t)\right)$. The log-likelihood is an uninterpretable unitless measure that we ideally want to maximize. Following Kégl et al. (2021), we normalize $\mathcal{L}$ with the log-likelihood of a multivariate unconditional Gaussian distribution ($\mathcal{L}_{\text{baseline}}$) whose parameters are estimated from the dataset $\mathcal{D}$.

$$\text{LR}(\mathcal{D}; \theta; j \in \{1, \ldots, d_s\}) = \frac{e^{\mathcal{L}(\mathcal{D};\theta;j\in\{1,\ldots,d_s\})}}{e^{\mathcal{L}_{\text{baseline}}(\mathcal{D};j\in\{1,\ldots,d_s\})}} \tag{3}$$

**OUTLIER RATE (OR)**: In practice, the log-likelihood estimator is dominated by out-of-distribution test points where the likelihood tends to zero. For this reason, we omit the data points that have a likelihood smaller or equal to $p_{\min} = 1.47 \times 10^{-6}$ when computing the $LR$. The OR metric is the proportion of data points that fall in this category. Formally:

$$\text{OR}(\mathcal{D}; \theta; j \in \{1, \ldots, d_s\}) = 1 - \frac{\left|\left\{(s_t, a_t, s_{t+1}) \in \mathcal{D} : p_\theta^j(s_{t+1}|s_t, a_t) > p_{\min}\right\}\right|}{N} \tag{4}$$

OR is between 0 and 1, the lower the better.

**CALIBRATEDNESS (KS)**: This metric is computed using the quantile (under the model distribution) of the ground truth values. Hypothetically, these quantiles are uniform if the error we make on the ground truth is a random variable distributed according to a Gaussian having the predicted standard deviation, a property we characterize as *calibratedness*. To assess this, we compute the Kolmogorov-Smirnov (KS) statistics. Formally, starting from the model cumulative distribution function (CDF) $F_\theta(s_{t+1}|s_t, a_t)$, we define the empirical CDF of the quantiles of ground truth values by $\mathcal{F}_{\theta,j}(x) = \frac{\left|\left\{(s_t, a_t, s_{t+1}) \in \mathcal{D} | F_\theta^j(s_{t+1}|s_t, a_t) \leq x\right\}\right|}{N}$ for $x \in [0, 1]$. We denote by $U(x)$ the CDF of the uniform distribution over the interval $[0, 1]$, and we define the KS statistics as the largest absolute difference between the two CDFs across the data set $\mathcal{D}$:

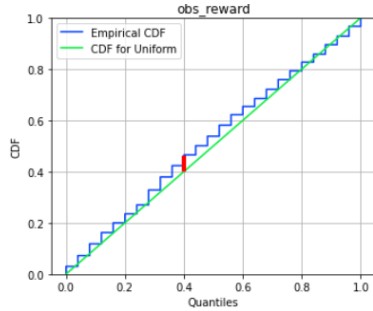

Figure 5: Kolmogorov-Smirnov (KS) statistic (in red) of the predicted reward.

$$\text{KS}(\mathcal{D}; \theta; j \in \{1, \dots, d_s\}) =$$

$$\max_{i \in \{1, \dots, N\}} \left| \mathcal{F}_{\theta,j}(F_\theta^j(s_{i,t+1}|s_{i,t}, a_{i,t})) - U(F_\theta^j(s_{i,t+1}|s_{i,t}, a_{i,t})) \right| \quad (5)$$

The KS score is between zero and one, the lower the better.

**LONG HORIZON METRICS KS($L$) AND R2($L$)**: Although the models are trained to optimize the one-step prediction log-likelihood score, we want to assess their precision and calibratedness at a longer horizon. Indeed, during the agent learning phase we sample trajectories of multiple steps which can lead to uncertain regions in the case of significant compounding errors down the horizon. For this purpose, we use ground truth actions from a system trace to generate a population of $n \in \mathbb{N}$ trajectories of length $L_{\max}$: $\mathcal{Y}_L = [\hat{s}_{\ell, t+1:t+L_{\max}}]_{\ell=1}^n$ and use the mean predictions to compute a Monte-Carlo estimate of the R2($L$) metric, for $L = 0, \dots, L_{\max}$, using the sample mean $\hat{\mu}_\theta(s_{t+L}|s_t, a_t) = \frac{1}{n}\sum_{\hat{s} \in \mathcal{Y}_L} \hat{s}_{t+L}$ as approximate prediction. For the KS($L$) metric, we estimate the model CDF with the order statistic $F_\theta(s_{t+L}|s_t, a_t) = \frac{|\{\hat{s} \in \mathcal{Y}_L : \hat{s}_{t+L} \leq s_{t+L}\}|}{n}$ among the population of trajectories.

## D   PER-DIMENSION STATIC METRICS

In all plots, as in Table 1, the KS score is multiplied by 1000, and the OR and R2 scores are multiplied by 10000,

## D.1 RANDOM DATASET

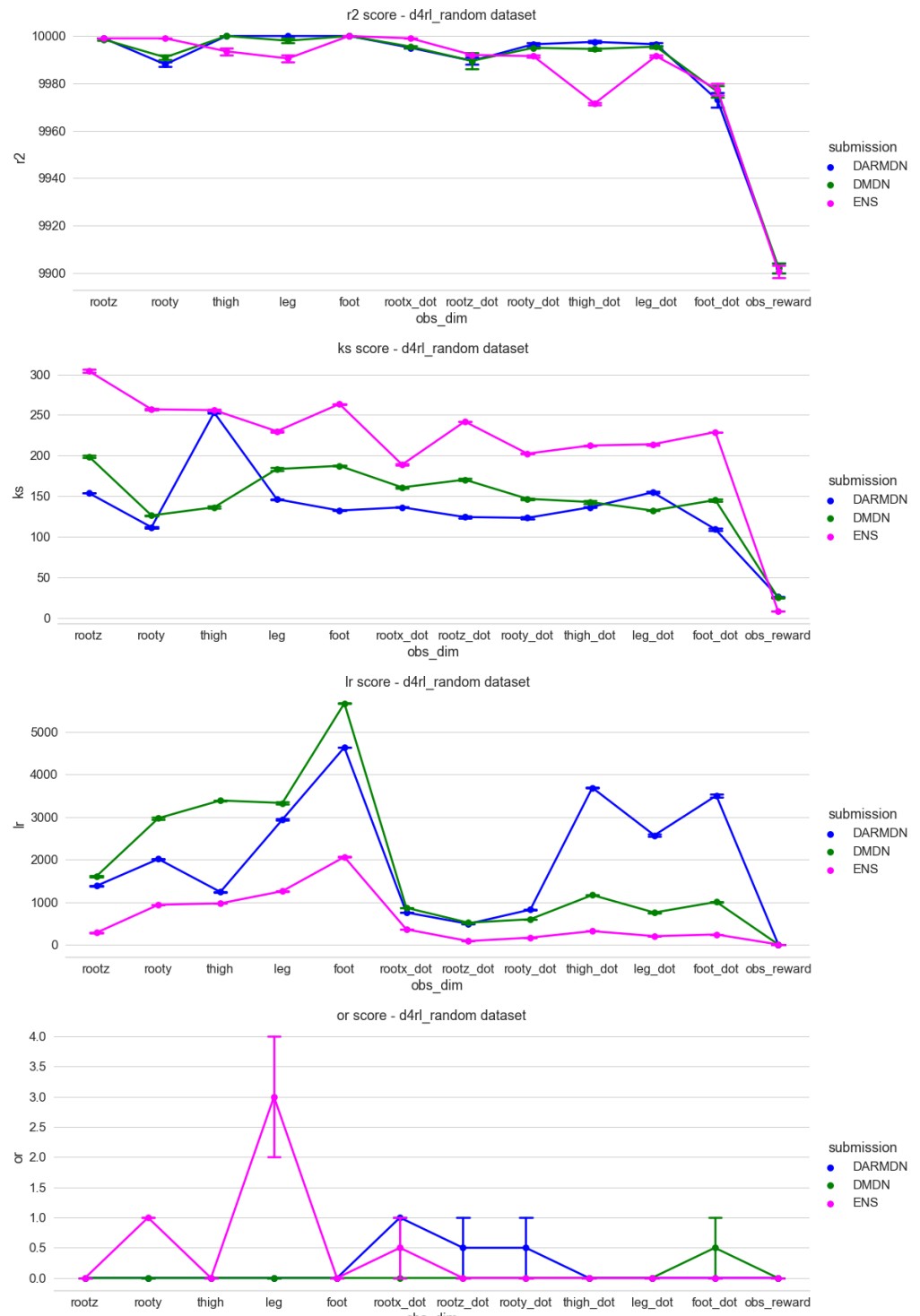

Figure 6: Per-dimension static metrics in the random dataset. The metrics include: **R2**, **KS**, **LR**, and **OR**. They are computed for all Hopper observables, in addition to the predicted reward (labeled *obs_reward*). The dots show the mean $\pm$ the standard deviation among the training and the validation scores for each metric.

### D.2 MEDIUM DATASET

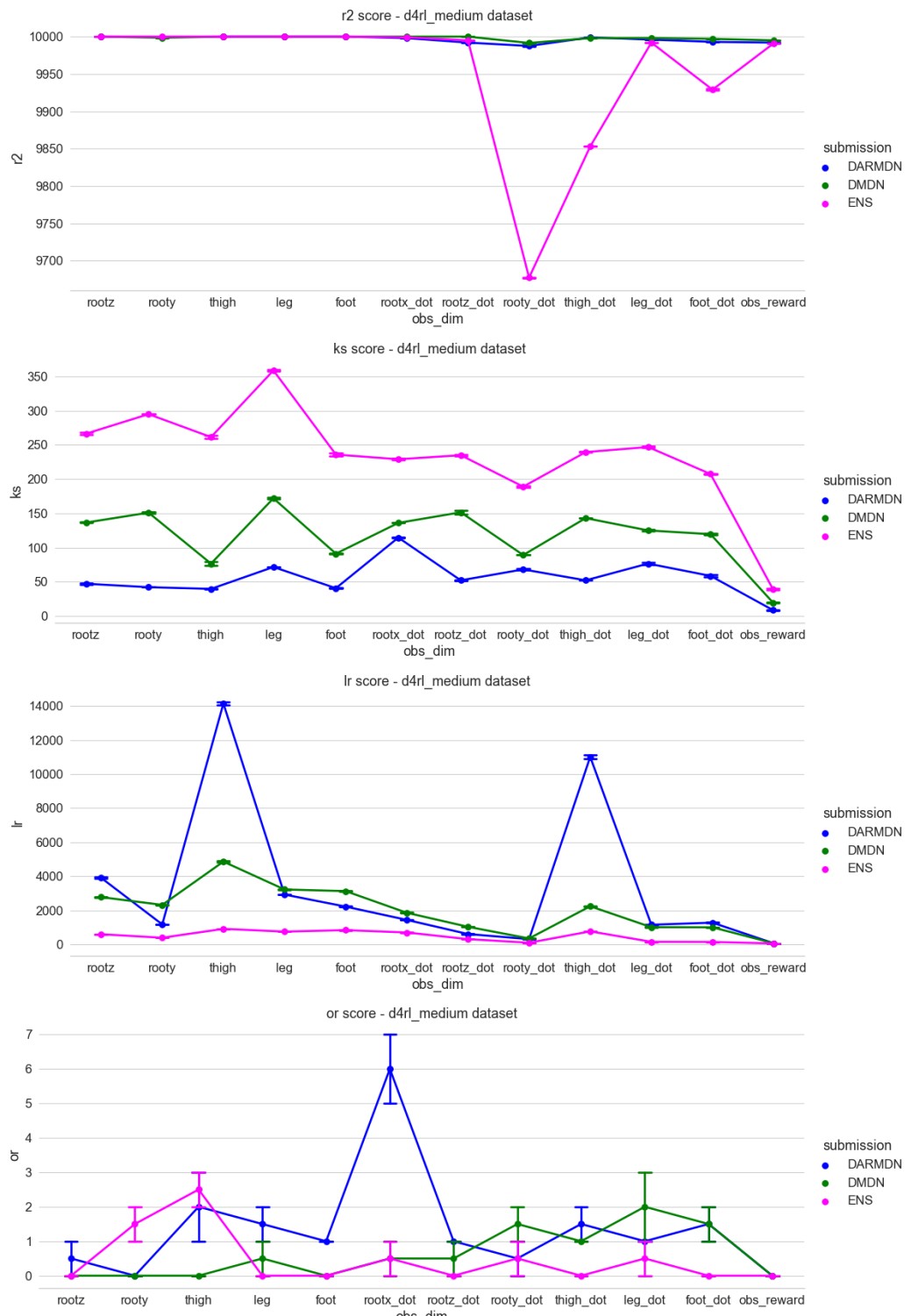

Figure 7: Per-dimension static metrics in the medium dataset. The metrics include: **R2**, **KS**, **LR**, and **OR**. They are computed for all Hopper observables, in addition to the predicted reward (labeled *obs_reward*). The dots show the mean ± the standard deviation among the training and the validation scores for each metric.

### D.3 MEDIUM-REPLAY DATASET

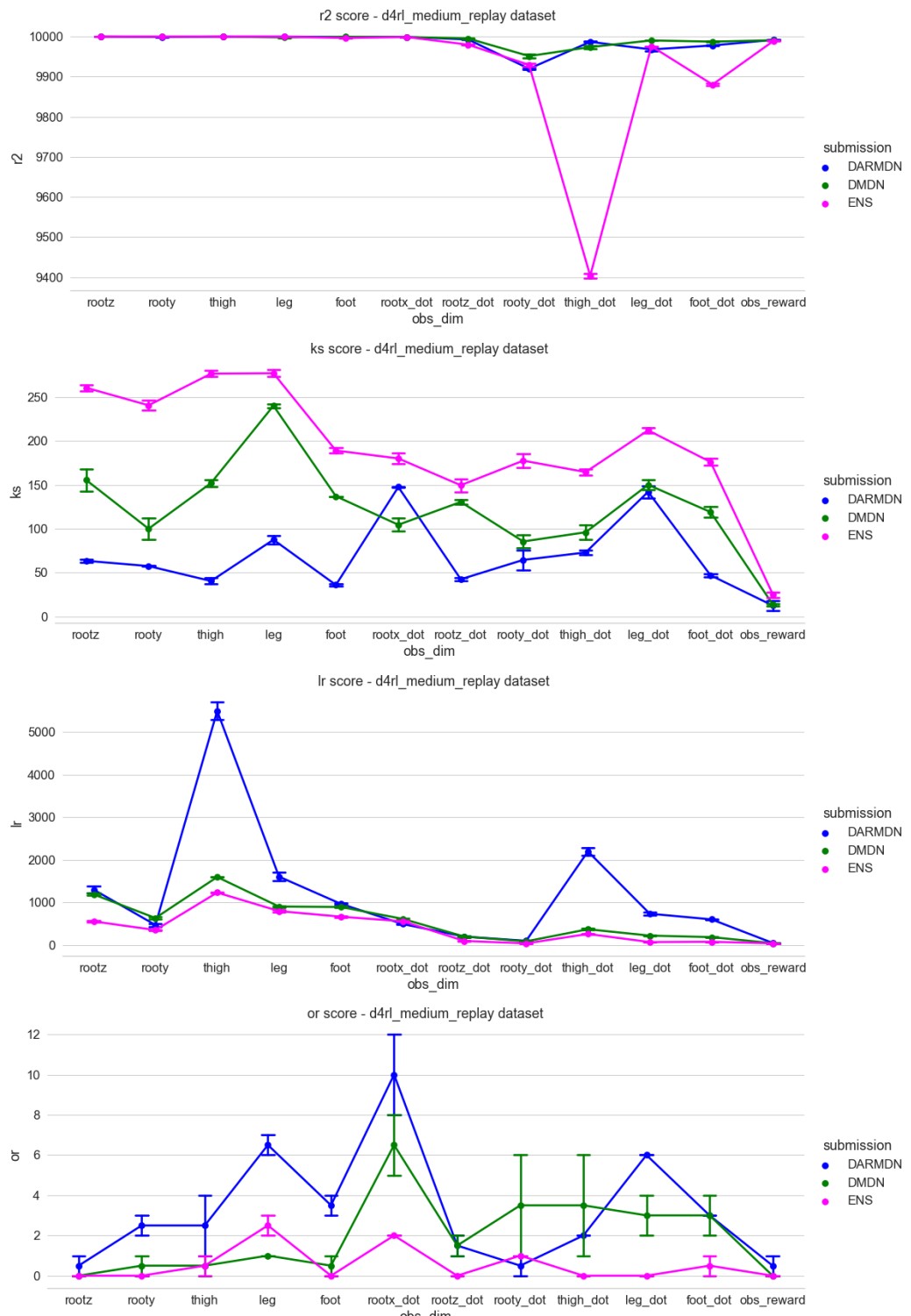

Figure 8: Per-dimension static metrics in the medium-replay dataset. The metrics include: **R2**, **KS**, **LR**, and **OR**. They and are computed for all Hopper observables, in addition to the predicted reward (labeled *obs_reward*). The dots show the mean ± the standard deviation among the training and the validation scores for each metric.

### D.4   MEDIUM-REPLAY DATASET

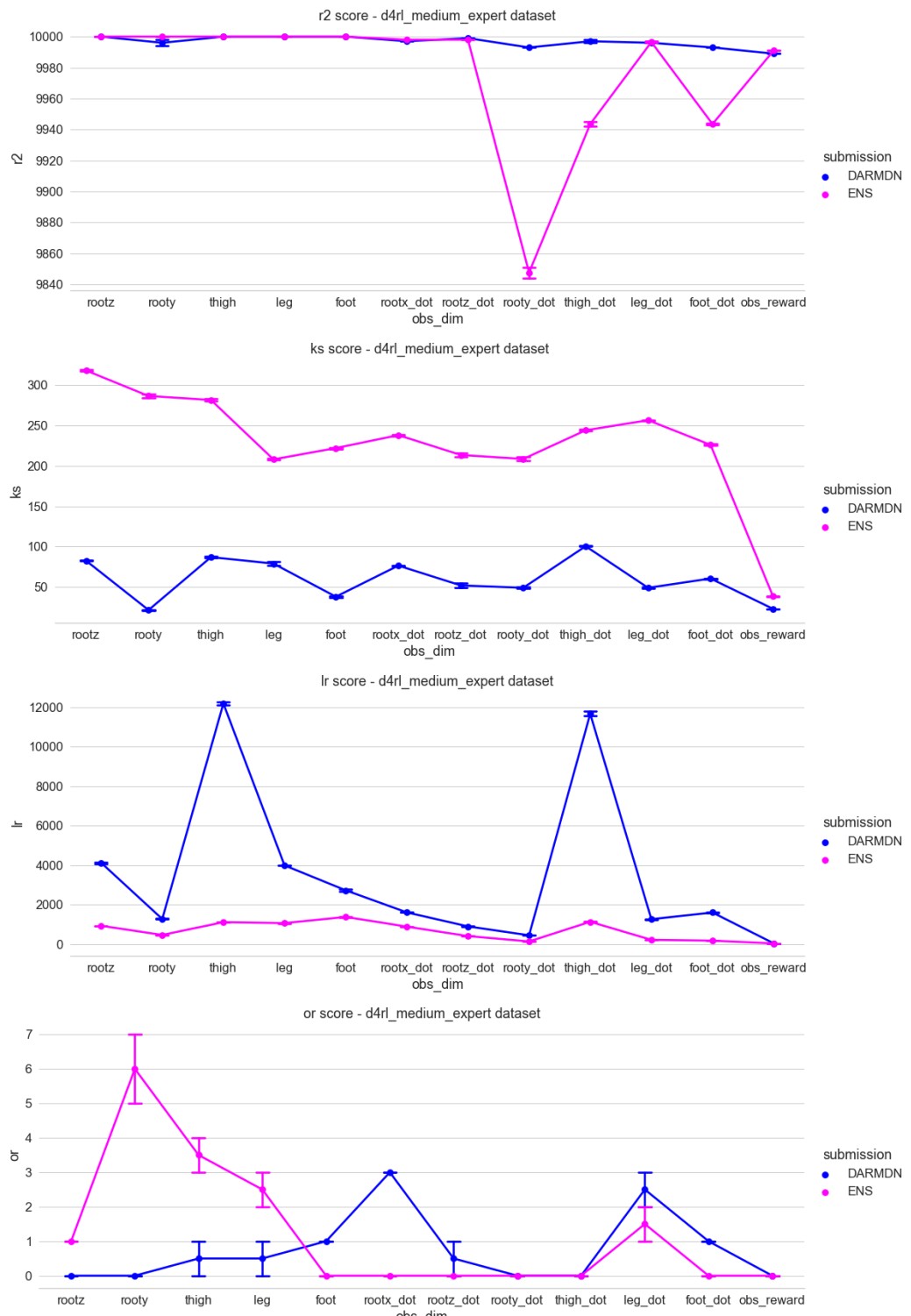

Figure 9: Per-dimension static metrics in the medium-expert dataset. The metrics include: **R2**, **KS**, **LR**, and **OR**. They are computed for all Hopper observables, in addition to the predicted reward (labeled *obs_reward*). The dots show the mean ± the standard deviation among the training and the validation scores for each metric.

# E    ERROR QUANTILE HISTOGRAMS

## E.1    RANDOM DATASET

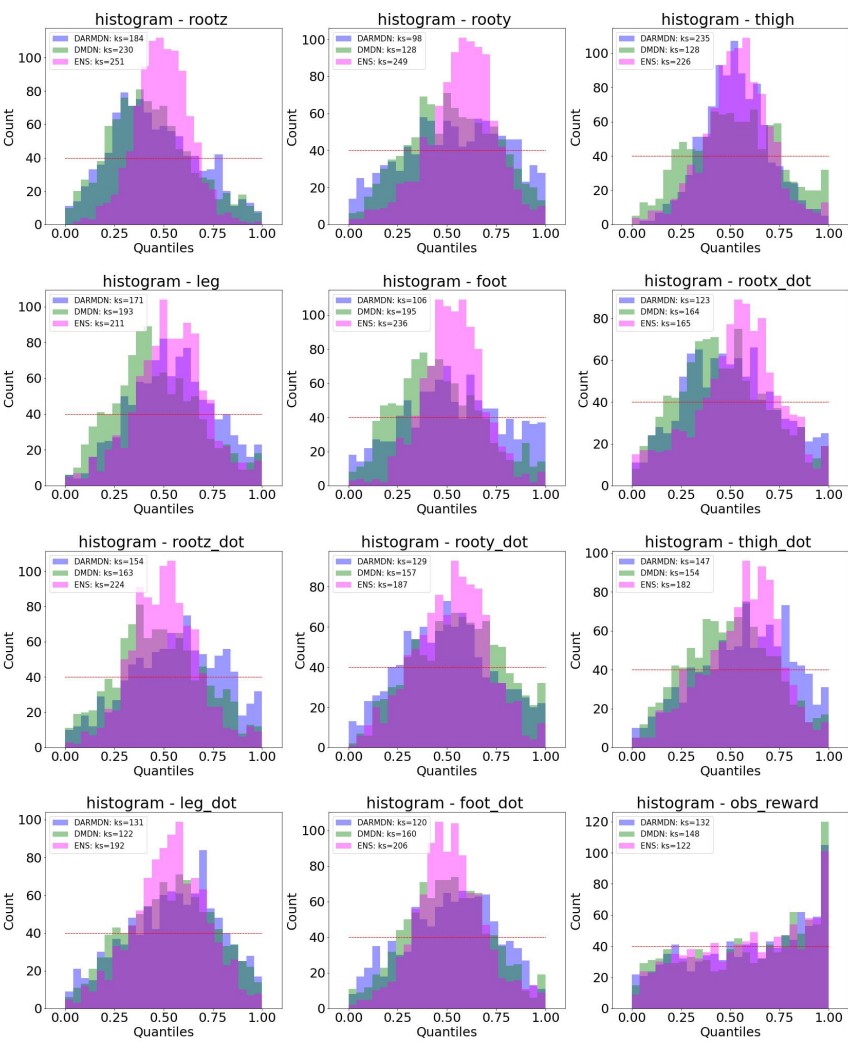

Figure 10: Per-dimension Error quantile histograms in the random dataset. The plot shows the ground truth validation quantiles under the model distribution. The legend includes the value of the KS calibratedness metric, and the dotted red line indicates the ideal case when the quantiles follow a uniform distribution. The histograms are computed for all Hopper observables, in addition to the predicted reward (labeled *obs_reward*).

## E.2 MEDIUM DATASET

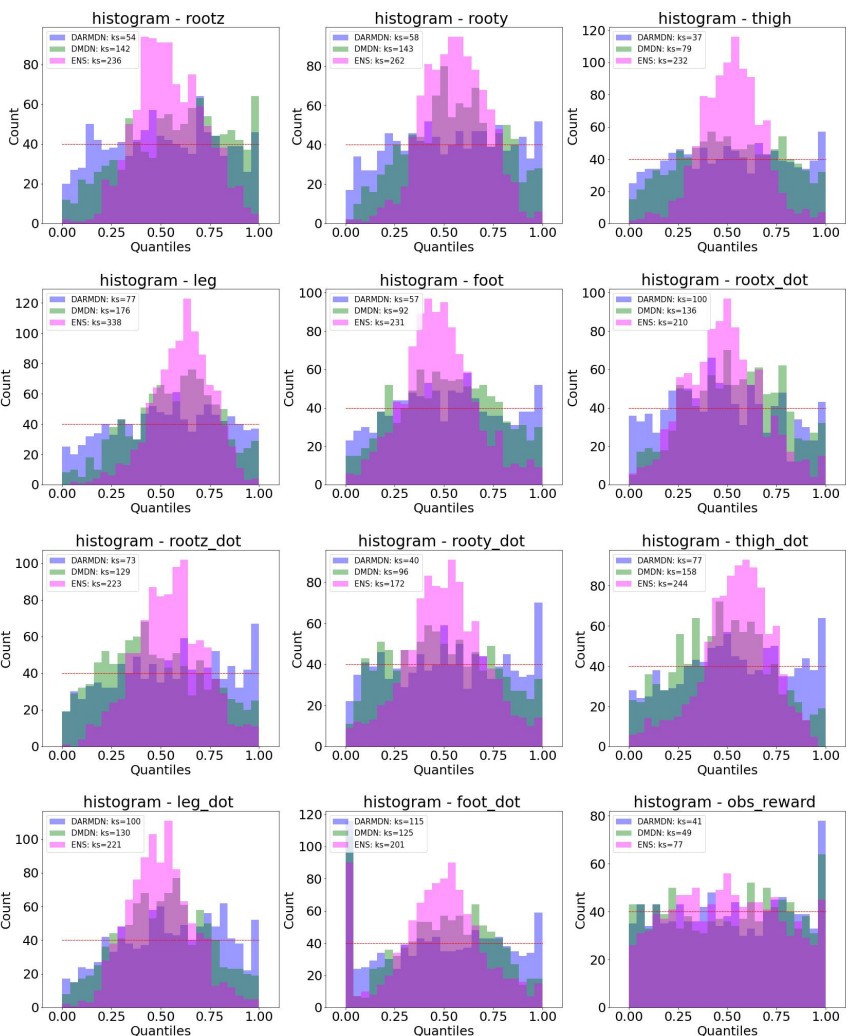

Figure 11: Per-dimension Error quantile histograms in the medium dataset. The plot shows the ground truth validation quantiles under the model distribution. The legend includes the value of the KS calibratedness metric, and the dotted red line indicates the ideal case when the quantiles follow a uniform distribution. The histograms are computed for all Hopper observables, in addition to the predicted reward (labeled *obs_reward*).

### E.3 MEDIUM-REPLAY DATASET

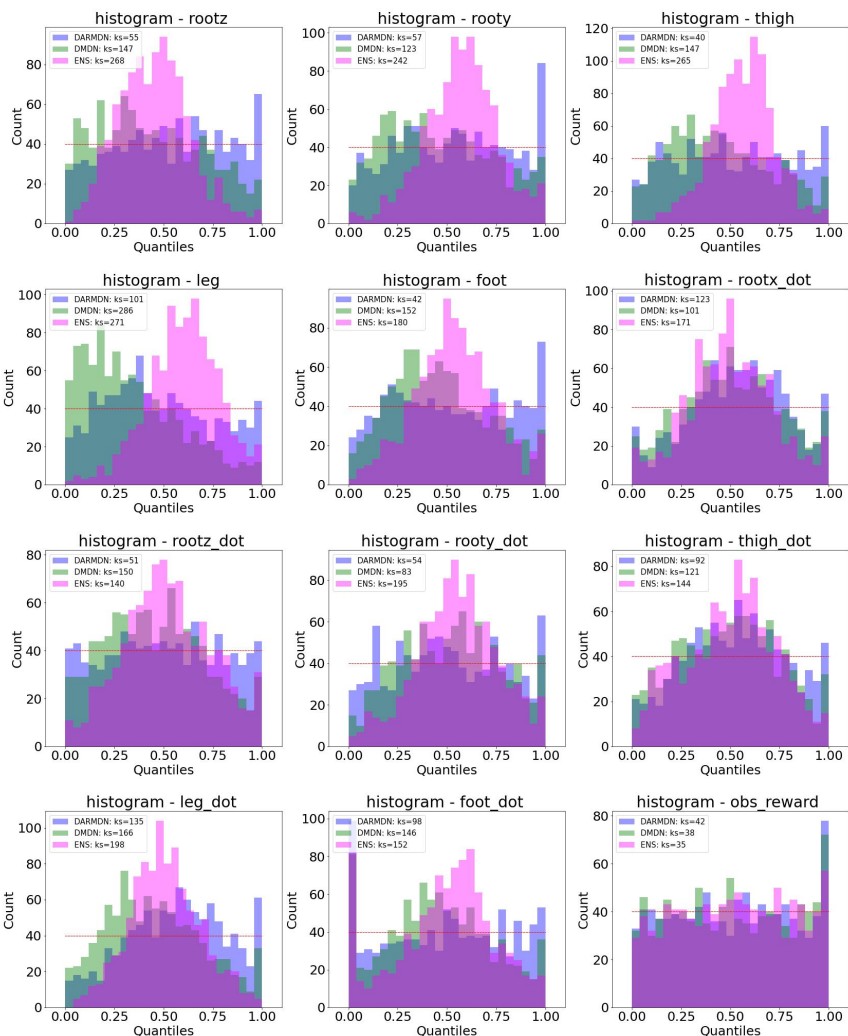

Figure 12: Per-dimension Error quantile histograms in the medium-replay dataset. The plot shows the ground truth validation quantiles under the model distribution. The legend includes the value of the KS calibratedness metric, and the dotted red line indicates the ideal case when the quantiles follow a uniform distribution. The histograms are computed for all Hopper observables, in addition to the predicted reward (labeled *obs_reward*).

### E.4 MEDIUM-EXPERT DATASET

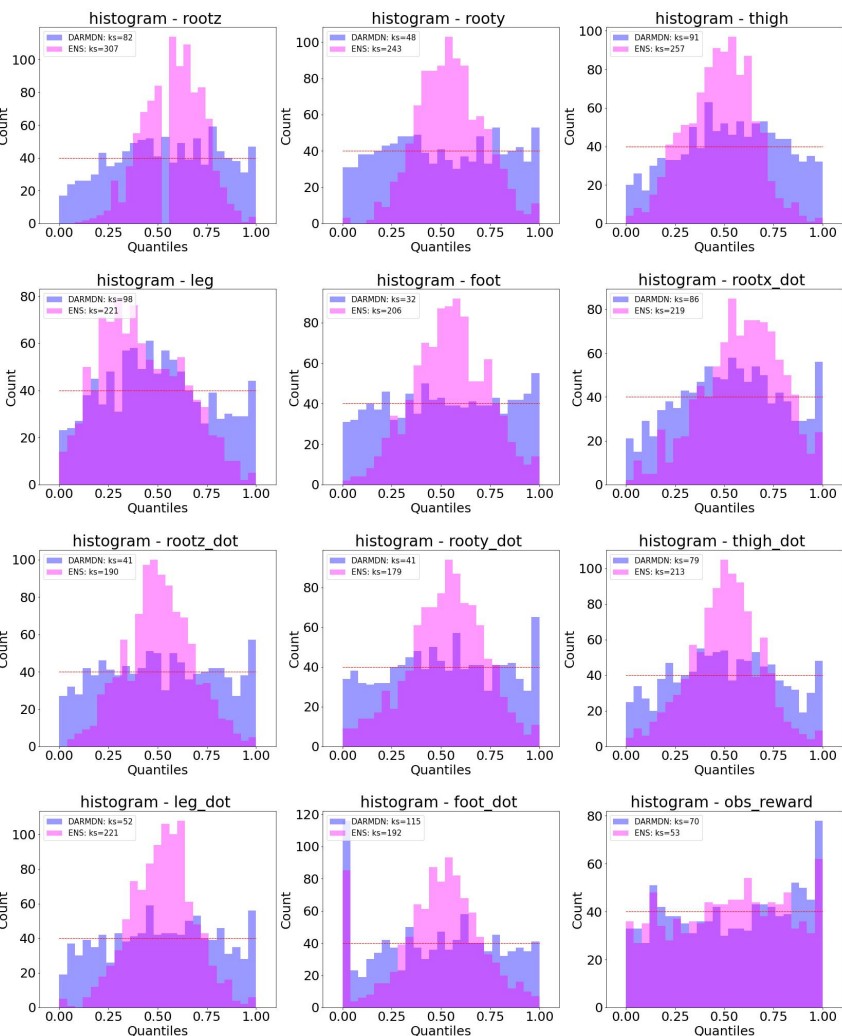

Figure 13: Per-dimension Error quantile histograms in the medium-expert dataset. The plot shows the ground truth validation quantiles under the model distribution. The legend includes the value of the KS calibratedness metric, and the dotted red line indicates the ideal case when the quantiles follow a uniform distribution. The histograms are computed for all Hopper observables, in addition to the predicted reward (labeled *obs_reward*).

# F LONG HORIZON METRICS

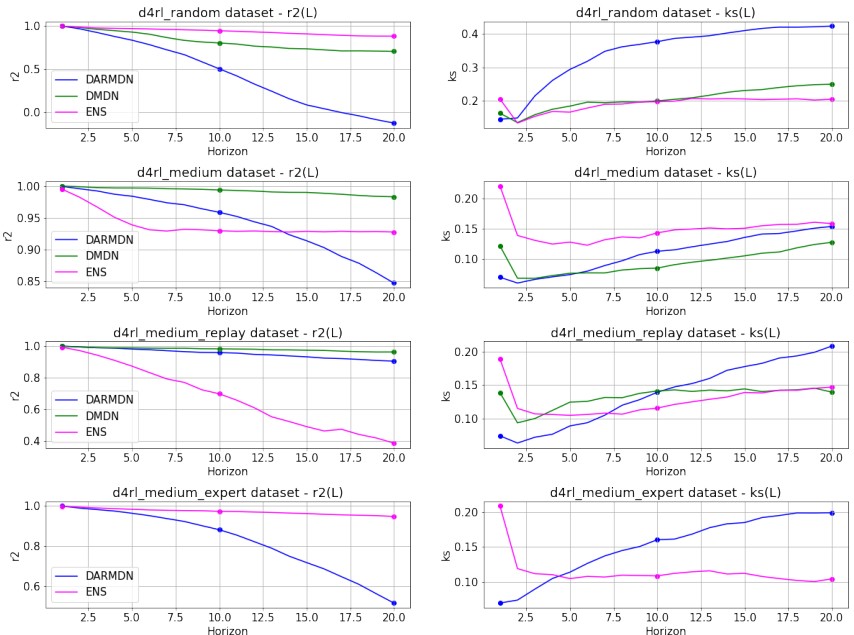

Figure 14: Long horizon explained variance **R2**($L$) and calibratedness **KS**($L$). The metric is aggregated by averaging over Hopper's observables and predicted reward.

# G  STATIC AND DYNAMIC METRICS CORRELATIONS

## G.1  RANDOM DATASET

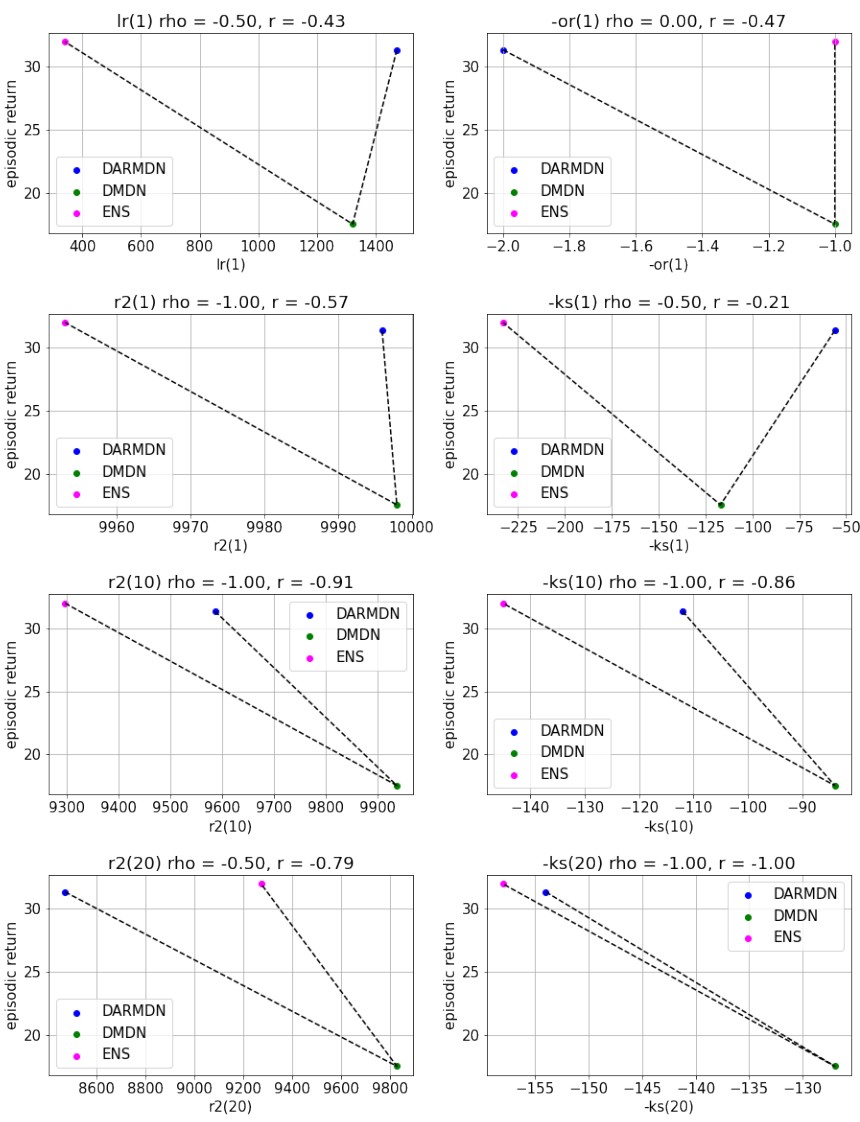

Figure 15: The Spearman and Pearson correlations between the episodic return and the static metrics (**LR**, negative **OR**, **R2**(1), negative **KS**(1), **R2**(10), negative **KS**(10), **R2**(20), negative **KS**(20)) in the random dataset. To uniformly evaluate the metrics' positive correlation with the episodic return, we take the negative of the metrics where the smaller is the better (**KS**($L$) and **OR**).

## G.2 MEDIUM DATASET

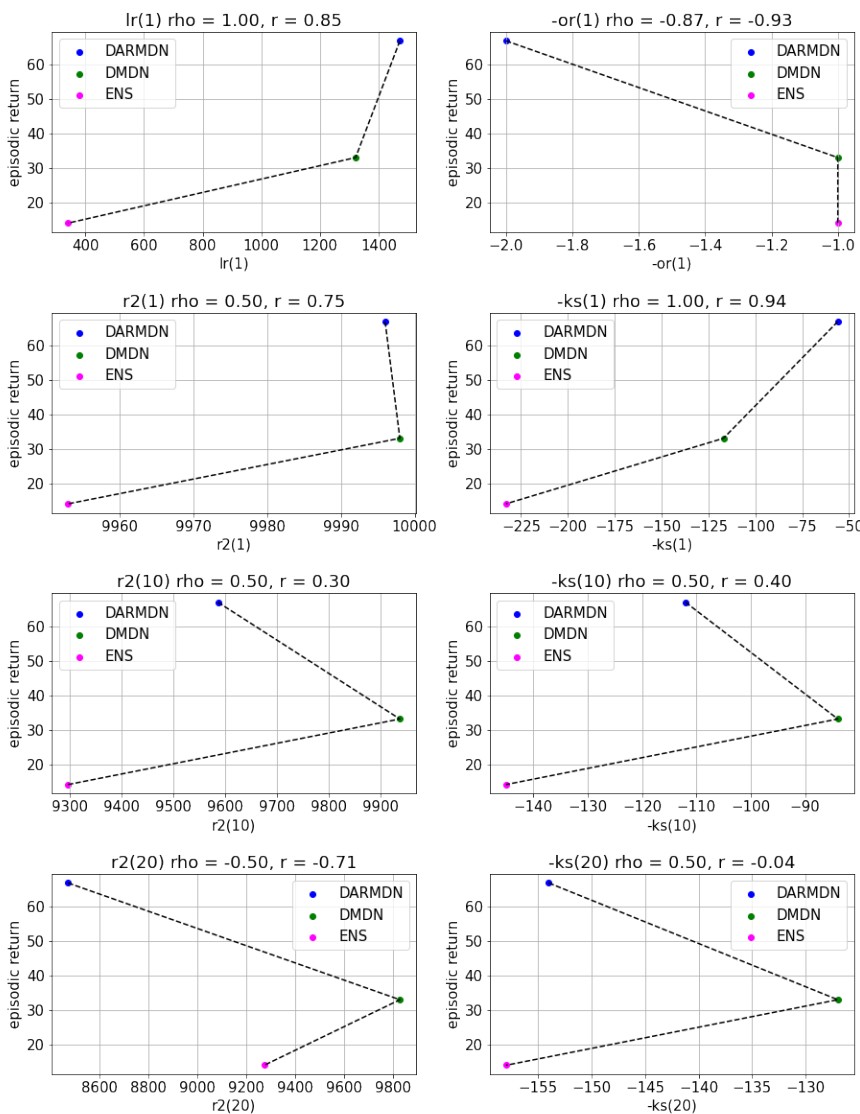

Figure 16: The Spearman and Pearson correlations between the episodic return and the static metrics (**LR**, negative **OR**, **R2**(1), negative **KS**(1), **R2**(10), negative **KS**(10), **R2**(20), negative **KS**(20)) in the medium dataset. To uniformly evaluate the metrics' positive correlation with the episodic return, we take the negative of the metrics where the smaller is the better (**KS**($L$) and **OR**).

### G.3 MEDIUM-REPLAY DATASET

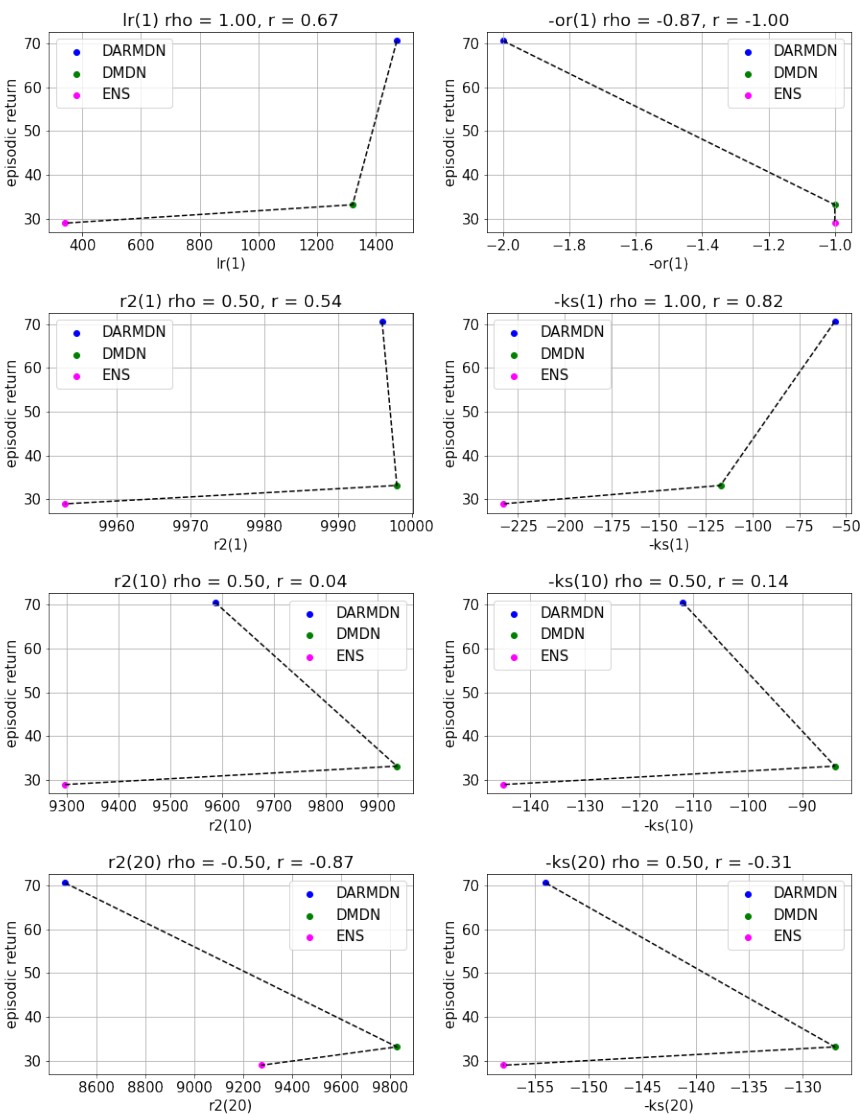

Figure 17: The Spearman and Pearson correlations between the episodic return and the static metrics (**LR**, negative **OR**, **R2**(1), negative **KS**(1), **R2**(10), negative **KS**(10), **R2**(20), negative **KS**(20)) in the medium-replay dataset. To uniformly evaluate the metrics' positive correlation with the episodic return, we take the negative of the metrics where the smaller is the better (**KS**($L$) and **OR**).

## G.4 MEDIUM-EXPERT DATASET

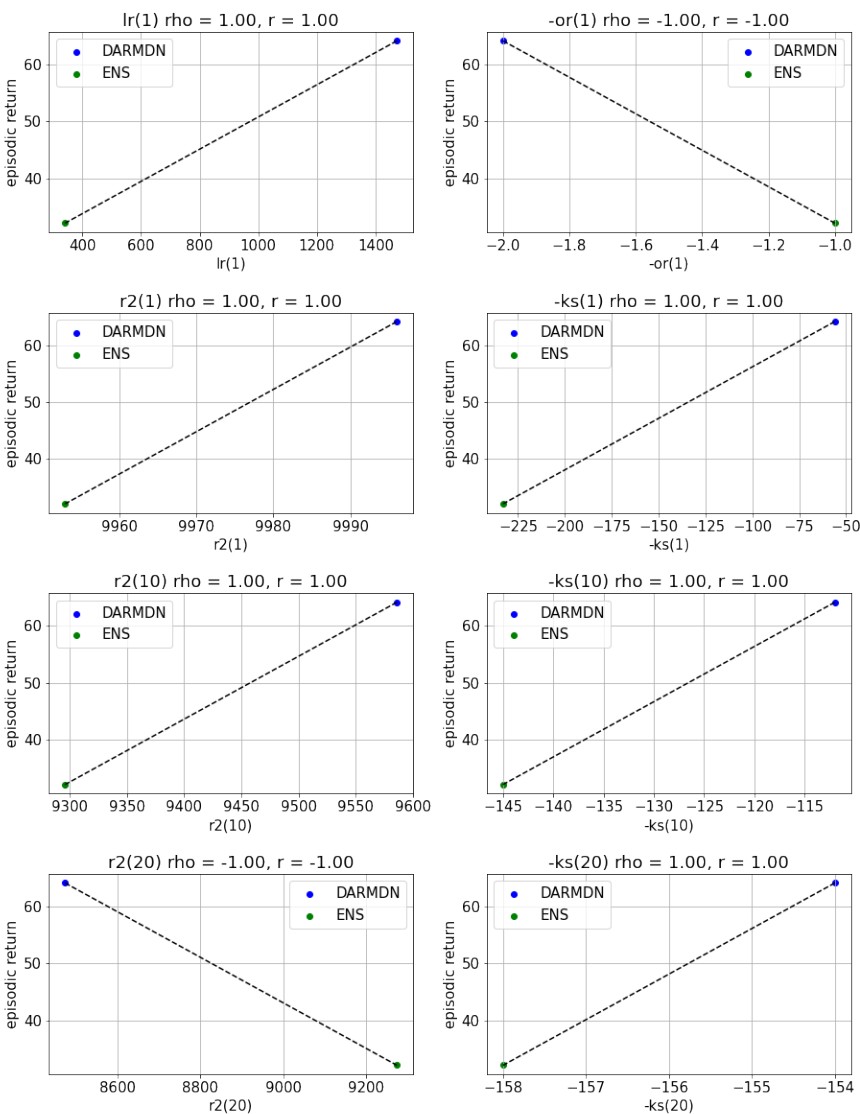

Figure 18: The Spearman and Pearson correlations between the episodic return and the static metrics (**LR**, negative **OR**, **R2**(1), negative **KS**(1), **R2**(10), negative **KS**(10), **R2**(20), negative **KS**(20)) in the medium-expert dataset. To uniformly evaluate the metrics' positive correlation with the episodic return, we take the negative of the metrics where the smaller is the better (**KS**($L$) and **OR**).

