# OpenReview forum: "Deep autoregressive density nets vs neural ensembles for model-based offline reinforcement learning"
_ICLR.cc/2023/Conference — Submitted to ICLR 2023_

### Official Review · Reviewer_eNAT · 2022-10-20

**Confidence:** 4
**Correctness:** 3
**Technical Novelty And Significance:** 3
**Empirical Novelty And Significance:** 3
**Recommendation:** 6

**Clarity, Quality, Novelty And Reproducibility:**

**Clarity** very good

**Quality** good

**Novelty** good

**Reproducibility** fair


**Strength And Weaknesses:**

**Strengths**
* The paper is well written.
* The paper makes an interesting contribution to the discussion of uncertainty quantification of dynamic models.

**Weaknesses**
* The empirical investigation relies on only one environment (Hopper).

**Further suggestions for improvement**
* The statement "Our algorithm achieves better performance than MOPO in all datasets" should be softened, e.g. "Our algorithm achieves equal or better performance than MOPO in all datasets", because in medium_replay the performance is not significantly better.

* In Table 2 and Table 3 standard deviations are given after $\pm$. Correctly, the $\pm$ sign is used to indicate the uncertainty of the measurement. So a confidence interval or a standard error.
This serves to ensure that the statistical significance of differences in the measured values can be easily grasped.

* Uncertainties should be stated with one or at most two valid digits.

* The number of decimal places in A $\pm$ B must match. E.g. "31.34 ± 0.5" -> "31.3 ± 0.5".

* Using only three repetitions (seeds) leads to unreliable results. If it is somehow possible, there should be more, e.g. 10 or 50.

* In "Other methods include behavior regularized policy optimization", also (Fujimoto et al., 2019) should be cited.

* Similar to the present paper, (Depeweg et al., Decomposition of uncertainty in Bayesian deep learning for efficient and risk-sensitive learning 2018) estimates uncertainty without ensemble and uses the uncertainty for conservatives in offline RL, it should, therefore, be cited. However, long roll-outs are used for the return estimation and no Q-function is used, so that the approach is structurally clearly different from MOPO.

* It is claimed "Conservatism in MBRL is achieved by uncertainty-based penalization of the model predictions." however, this is only one possible way, another possibility is also in model-based the behavior regularized policy optimization, e.g. (Swazinna et al, Overcoming model bias for robust offline deep reinforcement learning, 2021). Presumably other techniques exist, so it is probably better to write, e.g., "Conservatism in MBRL is frequently achieved by uncertainty-based penalization of the model predictions."

* It would be interesting to study the behavior in a stochastic environment, because only in stochastic environments the problem exists to separate aleatory and epistemic uncertainty (future work).


**Summary Of The Paper:**

The paper deals with uncertainty quantification of models in model based offline RL.
The paper uses an autoregressive dynamic model and shows that a single model that can provide an uncertainty estimate can be used as an alternative to uncertainty estimation using ensembles. Different methods are compared on the hopper benchmark from the D4RL benchmark collection.

**Summary Of The Review:**

The paper makes an interesting contribution to the discussion of uncertainty quantification of dynamic models.

---

> ### Author Response · Authors · 2022-11-18
> **Response**
>
> **1. The sign "+-" is used to indicate the uncertainty of the measurement. So a confidence interval or a standard error.**
>
> In table 2 and 3, we report the average score ± the standard deviation following MOPO (since it's our main source of comparison).
>
> **2. Using only three repetitions (seeds) leads to unreliable results. If it is somehow possible, there should be more, e.g. 10 or 50**
>
> Ideally we would run more than ten seeds to get robust performance metrics. However, we follow previous work in our choice of three seeds (as done in the Morel and the D4RL benchmark papers), and do not differ much from the rest of the algorithms we compare with (six seeds are used in MOPO and COMBO).
>
> **3. Try a stochastic environment where epistemic/aleatoric uncertainty separation would be of use (future work).**
>
> This is indeed part of our future work as engineering systems are noisy in most of the cases.
>
> **4. Alleviate the statement "Better than MOPO". Uncertainty notation +- same number of decimals. Some additional references and reformulation.**
>
> We modified the paper and took these suggestions into account.

---

> > ### Comment · Reviewer_eNAT · 2022-11-18
> > **It should not become the standard to repeat the flaws of others**
> >
> >  Using only three values to calculate the mean leads to high uncertainties. If it is possible, more repetitions should be used to reduce the uncertainty of the results. If this is not possible, justification must be given as to why it was not possible to use more repetitions.
> >
> > The established practice in natural science publications is that each measurement result is reported with its uncertainty. The $\pm$ sign is used exclusively to indicate the uncertainty. The standard deviation is not a consistent measure for calculating the uncertainty, because while the uncertainty becomes smaller the more repetitions are made, the standard deviation does not.
> >
> > It is very regrettable that in some recent papers, these elementary rules were disregarded, this should by no means become common practice.

---

> > > ### Author Response · Authors · 2022-11-19
> > > **We agree**
> > >
> > > Ah, we agree full-heartedly with you, and we will definitely work on this aspect in future resubmissions.
> > >
> > > What we face is a dilemma: either we do proper experiments with many seeds on smaller envs and smaller data, in which case reviewers tend to say we don't follow the protocol set up by the community, or we follow the partly flawed protocol that was set up, and face your rightful criticism. The problem is not only the number of seeds but also that the scores are highly non-Gaussian, so the standard deviations usually don't describe well what is happening (a certain number of seeds have catastrophically bad scores).
> > >
> > > We will definitely link your review in future resubmissions, thank you for your boldness!

---

> > > > ### Comment · Reviewer_eNAT · 2022-11-19
> > > > **What is important is whether the results are statistically significant**
> > > >
> > > > I think it would be a possibility to give the mean values $\pm$ uncertainty (here the standard error (i.e. stdev / sqrt( 3 - 1) ) ) for just 3 seeds. In three out of four cases the presented method seems to be statistically significant better than MOPO. And until the delivery of a camera ready version, perhaps experiments can still run, so that the results are based on a larger number of experiments.

---

> > > > > ### Author Response · Authors · 2022-11-19
> > > > > **We agree**
> > > > >
> > > > > We'll definitely do that in case the paper goes through, or in subsequent resubmissions.

---

### Official Review · Reviewer_ArJf · 2022-10-23

**Confidence:** 3
**Correctness:** 3
**Technical Novelty And Significance:** 3
**Empirical Novelty And Significance:** 3
**Recommendation:** 6

**Clarity, Quality, Novelty And Reproducibility:**

The ideas and arguments are clear. The quality of writing is good. The idea of using the predicted variance of an autoregressive model to penalize the policy is novel as far as I am aware. The authors say they will publish the code for reproducibility.

**Strength And Weaknesses:**

**strengths**

* The paper questions the use of ensembles in certain scenarios. Given the amount of attention that ensembles have received over the last years, the results of this paper are quite interesting.
* The experiments are in favor of the approach.
* Looking at different metrics and their correlation with RL performance is quite helpful and interesting.

**weaknesses**

* The experiments feature a single mujoco environment. This is my main reason for giving the paper a 6 over an 8.
* Autoregressive models are more computationally expensive than single neural nets and ensembles (though this might depend on the implementation).
* The autoregressive models are clearly outperformed when one considers L-step metrics. This would have ramifications about the applicability of the results. In works like PETS [1], predictive performance over a longer horizon is essential. The results in this paper suggest that autoregressive models are not a good fit for that. While I can understand that 1-step performance and L-step performance might not necessarily agree (e.g. one can sacrifice one to improve the other), it would be a little odd that one model category is better in the offline setting, while the other is better in the other.
* Given that the authors' initial pitch for autoregressive models is about correlations between state dimensions, I think the experiments section should check if there is indeed a difference between the two model classes in this regard. Perhaps a toy experiment involving a pendulum, where one could check the same sine-cosine scenarion that was described in the paper.

----

[1] Deep Reinforcement Learning in a Handful of Trials, https://arxiv.org/pdf/1805.12114.pdf

**Summary Of The Paper:**

This paper considers using autoregressive models to predict the different state dimensions when learning a transition model. The authors argue that autoregressive models are able to capture correlations between state dimensions and therefore work better than standard neural nets in the context of model-based offline RL. Experiments support these claims.

**Summary Of The Review:**

This paper proposes using autoregressive models for offline RL, arguing that these are better able to capture correlations between state dimensions. Experiments show a clear advantage of single neural nets and ensembles. The authors look at various performance metrics and check their correlation with final control performance. The contributions are worthy of acceptance, though the paper is weighed down by its narrow experimental scope. Repeating these experiments in a few more mujoco tasks would greatly strengthen this paper.

---

> ### Author Response · Authors · 2022-11-18
> **Response**
>
> **1. Autoregressive models are more computationally expensive than neural ensembles (depending on the implementations)**
>
> We agree that the actual computational cost of the considered models highly depends on the implementation and its optimization with respect to the infrastructure at disposal. In addition to this, another parameter in the equation is that fewer parameters are needed for a single dimension predictor compared to a multivariate one. This is stated in the hyperparameter optimization part (see Appendix A). For instance in the D4RL medium dataset, we found that a single hidden layer with 200 units was optimal for the autoregressive components, compared to three hidden layers with 500 units for the multivariate non-autoregressive model. Another possibility to further reduce the cost of autoregressive models is to separately tune the neural architecture for every component. This can lead to significantly smaller components in case of easily learnable functional dependencies.
>
> **2. Poor L-step metric for autoregressive models is a bit problematic (weird) in the sense that this makes them suitable for the offline setting but completely irrelevant in the iterated batch setting (reference to PETS where L-step metrics were discussed to be important for the final performance)**
>
> The setting considered in our paper differs from that of PETS in the fact that we use a single-step policy learning algorithm (soft-actor critic) while PETS use long-horizon decision-time planning. We believe that the important features for one setting do not necessarily transfer to the other.

---

### Official Review · Reviewer_xTqk · 2022-10-25

**Confidence:** 4
**Correctness:** 3
**Technical Novelty And Significance:** 2
**Empirical Novelty And Significance:** 1
**Recommendation:** 3

**Clarity, Quality, Novelty And Reproducibility:**

The paper is written quite clear The insights from the paper do not provide much novelty compared to previous work, mainly based on the limited evaluation. The comparison between models is interesting, but only done on a single benchmark.

**Strength And Weaknesses:**

Strengths:
- the use of auto-reggressive models and its comparison to  standard ensembling methods appears novel

Weaknesses:
- Significance : The methodology the authors present (offline model-based RL with a parameterized policy) is standard. The algorithms for model learning (deep ensembles, auto-regressive models) are known and testing on a single benchmark does not provide any insight into their advantages and disadvantages
- Limited experiments: The authors say in the conclusion "In this paper, we ask what are the best dynamic system models, estimating their own uncertainty,"  -- but all tests are done on a single benchmark.
- Limited comparisons: There is more work on using uncertainty-aware models in offline model-based RL, such as Gaussian Processes, Bayesian Deep Learning or using Dropout-mechanisms

**Summary Of The Paper:**

The paper uses a standard off-policy moder-based reinforcement learning algorithm and uses a set of dynamics models (autoregressive models, ensembles & mixture density models) on a single benchmark.

They then test the algorithms using a set of metrics and found that auto-regressive models appear to give improved performance



**Summary Of The Review:**

Overall, I cannot recommend acceptance because I unfortunately do not see a clear contribution and significance of the presented work.

---

> ### Author Response · Authors · 2022-11-18
> **Response**
>
> **1. Offline MBRL with parametrized policy is standard**
>
> The purpose of our paper is not to suggest a new methodology. We rather want to stress the fact that many aspects of the standard approach (MBRL with a parametrized policy) are still not yet fully understood, notably the respective impacts of the model and the agent on the return in the real system. Our contribution is to carefully study the influence of the model choice, and specifically if single autoregressive models can replace neural ensembles. We thus keep the rest of the standard approach unchanged (SAC agent, Surrogate MDP, uncertainty heuristics) as done in MOPO to set ourselves in a fair and comparable experimental setup.
>
> **2. The models used in the paper are well known/studied.**
>
> As far as we are aware few papers have considered autoregressive density nets in the context of MBRL, and yet, in these papers, the setting is different than ours (Iterative-batch Reinforcement Learning in Kegl et al. (2021), Offline Policy Evaluation in Zhang et al. (2021), and decision-time planning in Zhan et al. (2021)). In this work we challenge the default choice of neural ensembles with well-calibrated autoregressive models. Consequently, we designed experiments to study this very hypothesis.
>
> **3. Limited comparisons with other models that use uncertainty (GPs, BNNs, Dropout based ensembles)**
>
> We agree in the sense that a comparison between different models that use uncertainty is a relevant study that would give insight about uncertainty estimation while fixing the other components. However, this is not the goal of our study. We aim to compare neural networks based models (which are used to reach state-of-the-art performance in the literature) in order to suggest that autoregressive density nets can be used in place of the common neural ensembles. Furthermore, Kegl et al. (2021) did an in-depth comparison of some of the models mentioned in a growing batch setup, and found that density nets were the best performing and most reliable family of models. When allocating finite time and effort, we needed to make a decision where to put the experimental boundaries. But to improve our paper, we would like to ask the reviewer why he/she thinks those model families would be competitive on our setup.

---

### Author Response · Authors · 2022-11-18
**General comment**

We thank the reviewers for their comments on our paper. We first address a common concern here and then reply individually to each reviewer.

**Limited experiments (only one environment)**: We agree that more environments are needed in order to strengthen our findings. However, we would like to highlight the fact that we perform many experiments to assess different properties of the considered models. Our philosophy is to go deeper in one environment rather than to have a superficial yet large experimental scope. Furthermore, we consider four different datasets from the D4RL benchmark (which correspond to four different behavior policies, and consequently, four different problems). Hence, we believe that our experiments include enough diversity to back the legitimacy of our conclusions: neural ensembles are not the only approach to get good uncertainty estimates for model-based offline RL.

---

> ### Comment · Reviewer_eNAT · 2022-11-30
> **I agree**
>
> I agree that not using more environments should not really be a reason to reject the paper.

---

### Author Response · Authors · 2022-11-22
**Updated tables with 90% Gaussian confidence intervals instead of standard deviations.**

**1. Model dynamic evaluation: mean $\pm$ 90% Gaussian Confidence Interval over 3 seeds of the hyper-optimal SAC agents.**

| D4RL datasets | DARMDN | DMDN | ENS |
|---------------|:--------:|:------:|:-----:|
| random        | 31.34 $\pm$ 0.41  |  17.54 $\pm$ 7.96 |  31.97 $\pm$ 0.21 |
| medium        | 66.96 $\pm$ 5.14 |   33.12 $\pm$ 4.84 |  14.12 $\pm$ 10.76 |
| medium_replay | 70.57 $\pm$ 19.86 |  33.16 $\pm$ 1.54 |   28.96 $\pm$ 12.89 |
| medium_expert | 64.18 $\pm$ 20.39 |  \- $\pm$ -   |      32.10 $\pm$ 0.54 |

  **2. Results on the D4RL benchmark. The scores indicate the mean $\pm$
  90% Gaussian confidence interval across 3 seeds (6 seeds for MOPO)
  of the normalized episodic return. We take the scores of MBRL
  algorithms from their respective papers, and the scores of the model
  free algorithms and Behavior cloning (BC) from the D4RL paper
  [@Fu2021B].**

| D4RL datasets | BC | Ours | MOPO | COMBO | MOREL | SAC | BEAR | BRAC-v | CQL |
|---------------|:----:|:----:|:------:|:-------:|:-------:|:-----:|:------:|:--------:|:-----:|
  random   |  1.6  |   31.3 $\pm$ 0.4  |  11.7 $\pm$ 0.1  |  17.9  |  53.6  |  11.3 |  9.5 |   12.0  |   10.8 |
  medium    |      29.0 |   66.9 $\pm$ 5.1  |  28.0 $\pm$ 4.0  |  94.9  |  95.4 |  0.8  |  47.6 |  32.3  |   86.6 |
  medium_replay |  11.8  |  70.5 $\pm$ 19.8 |  67.5 $\pm$ 8.0 |   73.1  |  93.6  |  1.9  |  10.8 |  0.9   |   48.6 |
  medium_expert  | 111.9 |  64.1 $\pm$ 20.3 |  23.7 $\pm$ 1.95 |  111.1 |  108.7 |  1.6  |  4.0 |   0.8    |  111.0 |

---

### Decision · Program_Chairs · 2023-01-20

**Decision:**

Reject

**Justification For Why Not Higher Score:**

The paper could be considered for a poster, but the experiments do not properly quantify their uncertainty and thus should not be accepted.

**Justification For Why Not Lower Score:**

N/A

**Metareview: Summary, Strengths And Weaknesses:**

This paper investigates the effectiveness of autoregressive models versus ensemble models in estimating the transition dynamics and the uncertainty of the model in offline reinforcement learning. The paper's result shows that autoregressive models can be more effective than ensemble models for offline policy optimization. The reviewers note that the paper provides interesting results, delivering insights into how each model quantifies uncertainty and how model error correlates with optimized policy performance. They also note that some experiments need to be included to show why one model class can better model the dynamics or uncertainty than the other. Additionally, the reviewers point to the need for more than three trials to claim that one method is better. This paper has the potential for a solid contribution to the community. Still, it needs experiments to understand further why or when one model class will outperform the other and properly account for the performance metrics' uncertainty.

Side note: a deep investigation of a single environment is acceptable for publication. The important thing is to demonstrate new insights that are likely to generalize beyond that single environment.

**Summary Of Ac-Reviewer Meeting:**

N/A